# Percentile-based grain size distribution analysis tools (GSDtools) – estimating confidence limits and hypothesis tests for comparing two samples

Brett C. Eaton[1], R. Dan Moore[1], and Lucy G. MacKenzie[1]

[1]Geography, The University of British Columbia, 1984 West Mall, Vancouver, BC, Canada

**Correspondence:** Brett Eaton (brett.eaton@ubc.ca)

**Abstract.** Most studies of gravel bed rivers present at least one bed surface grain size distribution, but there is almost never any information provided about the uncertainty of the percentile estimates. We present a simple method for estimating the grain size confidence intervals about sample percentiles derived from standard Wolman or pebble count samples of bed surface texture. The width of a grain size confidence interval depends on the confidence level selected by the user (e.g., 95%), the number of stones sampled to generate the cumulative frequency distribution, and the shape of the frequency distribution itself. For a 95% confidence level, the computed confidence interval would include the true grain size parameter in 95 out of 100 trials, on average. The method presented here uses binomial theory to calculate a percentile confidence interval for each percentile of interest, then maps that confidence interval onto the cumulative frequency distribution of the sample in order to calculate the more useful grain size confidence interval. The validity of this approach is confirmed by comparing the predictions using binomial theory with estimates of the grain size confidence interval based on repeated sampling from a known population. We also developed a two-sample test of the equality of a given grain size percentile (e.g., $D_{50}$), which can be used to compare different sites, sampling methods or operators. The test can be applied with either individual or binned grain size data. These analyses are implemented in the freely available `GSDtools` package, written in the R language. A solution using the normal approximation to the binomial distribution is implemented in a spreadsheet that accompanies this paper. Applying our approach to various samples of grain size distributions in the field, we find that the standard sample size of 100 observations is typically associated with uncertainty estimates ranging from about $\pm 15\%$ to $\pm 30\%$, which may be unacceptably large for many applications. In comparison, a sample of 500 stones produces uncertainty estimates ranging from about $\pm 9\%$ to $\pm 18\%$. In order to help workers develop appropriate sampling approaches that produce the desired level of precision, we present simple equations that approximate the proportional uncertainty associated with the $50^{th}$ and $84^{th}$ percentiles of the distribution as a function of sample size and sorting coefficient; the true uncertainty of any sample depends on the shape of the sample distribution, and can only be accurately estimated once the sample has been collected.

## 1 Introduction

A common task in geomorphology is to estimate one or more percentiles of a particle size distribution, denoted $D_P$, where $D$ represents the particle diameter (mm) and the subscript $P$ indicates the percentile of interest. Such estimates are typically

used in calculations of flow resistance, sediment transport, and channel stability; they are also used to track changes in bed condition over time, and to compare one site to another. In fluvial geomorphology, commonly used percentiles include $D_{50}$ (which is the median) and $D_{84}$. In practice, sampling uncertainty for the estimated grain sizes is almost never considered during data analysis and interpretation. This paper presents a simple approach based on binomial theory for calculating grain
size confidence intervals, and for testing whether or not the grain size percentiles from two samples are statistically different.

Various methods for measuring bed surface sediment texture have been reviewed by previous researchers (Church et al., 1987; Bunte and Abt, 2001b; Kondolf et al., 2003). While some approaches have focused on using semi-qualitative approaches such as facies mapping (e.g. Buffington and Montgomery, 1999), or visual estimation procedures (e.g. Latulippe et al., 2001), the most common means of characterizing the texture of a gravel bed surface is still the cumulative frequency analysis of some
version of the pebble count (Wolman, 1954; Leopold, 1970; Kondolf and Li, 1992; Bunte and Abt, 2001a). Pebble counts are sometimes completed by using a random walk approach, wherein the operator walks along the bed of the river, sampling those stones that are under the toe of each boot and recording the b-axis diameter. In other cases, a regular grid is superimposed upon the sedimentological unit to be sampled, and the b-axis diameter of all the particles under each vertex is measured. In still other cases, computer-based photographic analysis identifies the b-axis of all particles in an image of the bed surface, though
this introduces potential uncertainties associated with how much of the particle is buried beneath the surface, and how much is visible from a photograph of the surface. Data are typically reported as cumulative grain size distributions for $0.5\phi$ size intervals (e.g., 8 - 11.3 mm, 11.3 to 16 mm, 16 - 22.6 mm, 22.6 - 32 mm, and so on), from which the grain sizes corresponding to various percentiles are extracted.

Operator error and the technique used to randomly select bed particles have frequently been identified as important sources
of uncertainty in bed surface samples (Hey and Thorne, 1983; Marcus et al., 1995; Olsen et al., 2005; Bunte et al., 2009), but one of the largest sources of uncertainty in many cases is likely to be associated with sample size, particularly for standard pebble counts of about 100 stones. Unfortunately, the magnitude of the confidence interval bounding an estimated grain size is seldom calculated and/or reported, and the implications of this uncertainty are – we believe – generally under-appreciated. To address this issue, we believe that it should become standard practice to calculate and graphically present the confidence
intervals about surface grain size distributions.

For the most part, attempts to characterize the uncertainty of pebble counts have focused on estimating the uncertainty of $D_{50}$, and have typically assumed that the underlying distribution is log normal (Hey and Thorne, 1983; Church et al., 1987; Bunte and Abt, 2001b); when used to determine the number of measurements required to reach a given level of sample precision, these approaches also require that the standard deviation of the underlying distribution be known, beforehand.
Attempts to characterize the uncertainty associated with other percentiles besides the median have relied on empirical analysis of extensive field data sets (Marcus et al., 1995; Rice and Church, 1996; Green, 2003; Olsen et al., 2005); however the statistical justification for applying those results to pebble counts from other gravel bed rivers having a different population of grain sizes is weak. Perhaps because of the complexity involved in extending the grain size confidence intervals about the median to the rest of the distribution, researchers almost never present confidence intervals on cumulative frequency distribu-
tion plots, or constrain comparisons of one distribution to another by any estimate of statistical significance. While others have

recognized the limitations of relatively small sample sizes (Hey and Thorne, 1983; Rice and Church, 1996; Petrie and Diplas, 2000; Bunte and Abt, 2001b), it still seems to be standard practice to rely on surface samples of about 100 observations.

Fripp and Diplas (1993) presented a means of generating confidence intervals bounding a grain size distribution. They presented a method for determining the minimum sample size required to achieve a desired level of sample precision using the normal approximation to the binomial distribution, wherein uncertainty is expressed in terms of the percentile being estimated (i.e., they estimated the percentile confidence interval), but not in terms of actual grain sizes (i.e., the grain size confidence interval). Based on their analysis, Fripp and Diplas (1993) recommended surface samples of between 200 and 400 stones to achieve reasonably precise results. Petrie and Diplas (2000) demonstrated that the percentile confidence interval predicted by Fripp and Diplas (1993) is similar to the empirical estimates produced by Rice and Church (1996), who repeatedly sub-sampled a known population of grain size measurements in order to quantify the confidence interval; Petrie and Diplas (2000) also recommended plotting the confidence intervals on the standard cumulative distribution plots as an easy way of visualizing the implications of sampling uncertainty. It is worth noting that the previous analyses were used to determine the sample size necessary to achieve a given level of sample precision; they were not adapted to the analysis and interpretation of previously collected surface distribution samples.

A number of studies have compared grain size distributions for two or more samples to assess differences among sites, sampling methods or operators (Hey and Thorne, 1983; Marcus et al., 1995; Bunte and Abt, 2001a; Olsen et al., 2005; Bunte et al., 2009; Daniels and McCusker, 2010). A simple approach would be to construct confidence intervals for the two estimates. If the confidence intervals do not overlap, one can conclude that the estimates are significantly different at the confidence level used to compute the intervals (e.g., 95%); and if a percentile estimate from one sample falls within the confidence interval for the other sample, then one cannot reject the null hypothesis that the percentile values are the same. However, the conclusion is ambiguous when the confidence intervals overlap but do not include both estimates; even for populations with significantly different percentile values, it is possible for the confidence intervals to overlap. Therefore, there is a need for a method to allow two-sample hypothesis tests of the equality of percentile values.

The objective of this note is to introduce robust, distribution-free approaches to (a) computing percentile confidence intervals and then mapping them onto a given cumulative frequency distribution from a standard pebble count in order to estimate the grain size confidence interval for the sample, and (b) conducting two-sample hypothesis tests of the equality of grain size percentile values. The approaches can be applied not only in cases in which individual grain diameters are measured, but also to the common situation in which grain diameters are recorded within phi-based classes, so long as the number of stones sampled to derive the cumulative distribution is also known.

The primary purpose of this work is to guide the analysis and interpretation of the grain size samples. While grain size confidence intervals are most applicable when comparing two samples to ascertain whether or not they are statistically different, we also demonstrate how knowledge of grain size uncertainty could be applied in a management context, where flood return period is linked to channel instability (for example). As we demonstrate in the paper, percentile uncertainty is distribution-free, and can be estimated using standard look-up tables similar to those used for t-tests, or using the normal approximation to the binomial distribution referred to by Fripp and Diplas (1993) (see Appendix A). Translating percentile

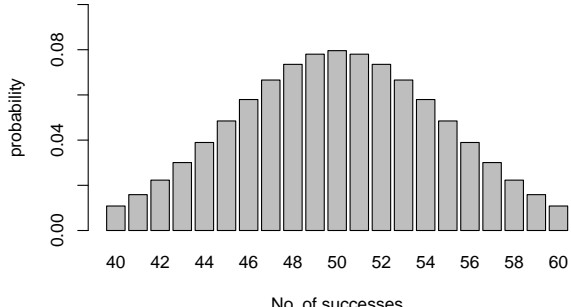

**Figure 1.** Binomial probability distribution for obtaining between 40 and 60 successes in 100 trials when the probability of success is 0.5. The probabilities for each outcome are calculated using Eq.1.

confidence intervals to grain size confidence intervals requires information about the grain size distribution, but is essentially a mapping exercise, not a statistical one. We implement both the estimation of a percentile confidence interval and the mapping of it onto a grain size confidence interval using: (1) a spreadsheet that we provide which uses the normal approximation to the binomial distribution, described by Fripp and Diplas (1993); and (2) an R package called `GSDtools` that we have written for this purpose that uses the statistical approach described in this paper. A demonstration is available online at `https://bceaton.github.io/GSDtools_demo_2019.nb.html`, which provides instructions for installing and using the `GSDtools` package; the demonstration is also included in the data repository associated with this paper. Finally, we use both existing data sets and the results from a Monte Carlo simulation to develop recommendations regarding the sample sizes required to achieve a pre-determined precision for estimates of the $D_{50}$ and the $D_{84}$.

## 2 Calculating confidence intervals

### 2.1 Overview

The key to our approach is that the estimation of any grain size percentile can be treated as a binomial experiment, much like predicting the outcome of a coin-flipping experiment. For example, we could toss a coin 100 times and count the number of times the coin lands head-side up. For each toss (of a fair coin, at least), the probability ($p$) of obtaining a head is 0.50. The number of times that we get heads during repeated experiments comprising 100 coin tosses will vary about a mean value of 50, following the binomial distribution (see Fig. 1).

The probability of getting a specific number of heads ($B_k$) can be computed from the binomial distribution:

$$B_k(k,n,p) = p^k(1-p)^{n-k}\frac{n!}{k!(n-k)!} \tag{1}$$

for which $k$ is the number successes (in this case, the number of heads) observed during $n$ trials for which the probability of success is $p$. The probabilities of obtaining between 40 and 60 heads calculated using Eq. 1 are shown in Fig. 1. The sum of all the probabilities shown in the figure is 0.96, which represents the coverage probability, $P_c$, associated with the interval from 40 to 60 successes.

We can apply this approach to a bed surface grain size sample based on a grid-based sampling approach or the standard Wolman pebble count approach, because for both methods the probability of sampling a stone of a given size is proportional to the relative area of the bed covered by stones of that size. Imagine that we are sampling a population of surface sediment sizes like that shown in Fig. 2a, for which the true median grain size of the population ($D_{50}$) is known (the population shown is defined by 3411 measurements of bed surface b-axis diameters at randomly selected locations in the wetted channel of a laboratory experiment performed by the authors, and has a median surface size of 1.7 mm). We know that half of the stream bed (by area) would be covered by particles smaller than the $D_{50}$, so for each stone that we select, the probability of it being smaller than the $D_{50}$ is 0.50. If we measure 100 stones and compare them to the $D_{50}$, then binomial sampling theory tells us that the probability of selecting exactly 50 stones that are less than $D_{50}$ is just 0.08, but that the probability of selecting between 40 and 60 stones less than $D_{50}$ is 0.96 (see Fig. 1).

Figure 2b shows a random sample of 100 stones taken from the population shown in Fig. 2a. Each circle represents a measured b-axis diameter, and all 100 measurements are plotted as a cumulative frequency distribution; the median surface size of the sample, $d_{50}$, is 1.5 mm. Note that, in this paper, we use $D_i$ to refer to grain size of the $i^{th}$ percentile for a population, and $d_i$ to refer to the grain size of the $i^{th}$ percentile of a sample. There are clear differences between the distribution of the sample and the underlying population, which is to be expected.

The first step in calculating a grain size confidence interval that is likely to contain the true median value of the population is to choose a confidence level; in this example, we set the confidence level to 0.96, corresponding to the coverage probability shown in Fig. 1. Looking at the upper boundary of binomial disribution in Fig. 1, an outcome in which 60 stones are smaller than $D_{50}$, corresponds to $d_{60} \leq D_{50}$; at the lower boundary, a result in which 40 stones are smaller than $D_{50}$ corresponds to $d_{40} \leq D_{50}$. As a result, the true value of the $D_{50}$ will fall between the sample $d_{40}$ and the sample $d_{60}$ 96% of the time under repeated random sampling. This represents the *percentile confidence interval* (see Fig. 2c), and it does not depend on the shape of the grain size distribution. For reference, a set of percentile confidence interval calculations are presented in Appendix B.

Once a confidence level has been chosen and the percentile confidence interval has been identified, a *grain size confidence interval* can be estimated by mapping the percentile confidence interval onto the sampled grain size distribution, as indicated graphically in Fig. 2d. Unlike the percentile confidence interval, the grain size confidence interval depends on the shape of the cumulative frequency distribution, and can only be calculated once the sample has been collected.

The approach demonstrated above for the median size can be applied to all other grain size percentiles by varying the probability $p$ in Eq. 1, accordingly. For example, the probability of picking up a stone smaller than the true $D_{84}$ of a population is 0.84, while the probability of picking up a stone smaller than the true $D_{16}$ is just 0.16. If we define $P$ to be the percentile of interest for the population being sampled, then the probability of selecting a stone smaller than that percentile is $p = P/100$, meaning that there is a direct correspondence between the grain size percentile and the probability of encountering a grain

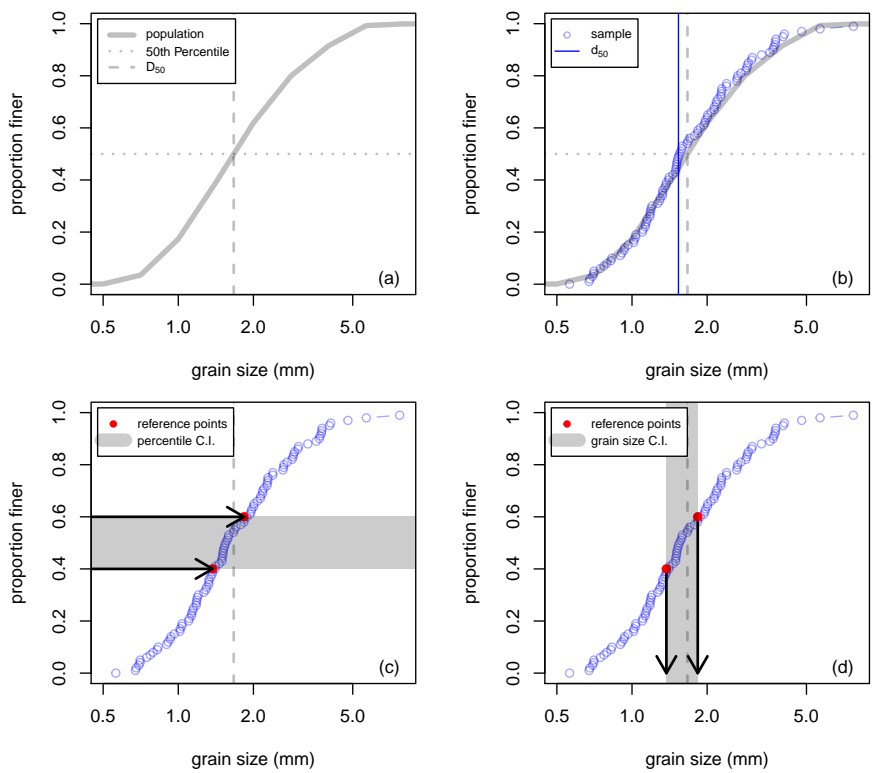

**Figure 2.** Defining the relation between the percentile confidence interval and the grain size confidence interval for a sampled $d_{50}$ value. (a) Begin with the known distribution for the population being sampled, with a vertical line indicating the true $D_{50}$. (b) Derive a sample distribution from 100 measurements from the population shown in (a) (note that the sample $d_{50}$ and the population $D_{50}$ are different). (c) Use binomial theory to estimate the percentile confidence interval that contains the population $D_{50}$. (d) Map the percentile confidence interval onto the sample cumulative frequency distribution to estimate the grain size confidence interval around the sample estimate, $d_{50}$ (note that the confidence interval does indeed contain the true $D_{50}$ for the population).

smaller than that percentile. As we show in the next section, the binomial distribution can be used to derive grain size confidence intervals for any estimate of $d_P$ for a sample that can be expected to contain the true value of $D_P$ for the entire population.

### 2.1.1 Statistical basis

This section presents the statistical basis for the approach outlined above in a more rigorous manner. In order to illustrate our approach for estimating confidence intervals in detail, we will use a sample of 200 measurements of b-axis diameters from our laboratory population of 3411 observations. These data are sorted in rank order and then used to compute the quantiles of the sample distribution. The difference between the cumulative distribution of raw data (based on 200 measurements of b-axis diameters) and the standard $0.5\phi$ binned data (which is typical for most field samples) is illustrated in Figure 3. While the calculated $d_{84}$ value for the binned data shown in Fig. 3a is not identical to that from the original data, the difference is small

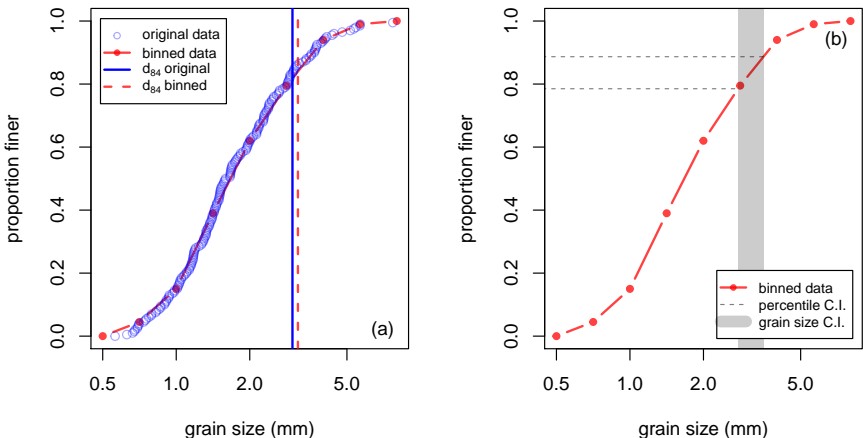

**Figure 3.** A grain size distribution from a stream table experiment based on a sample size of 200 observations. In Panel (a), blue circles indicate individual grain size measurements ($d_{(i)}$), and the red line is the cumulative frequency distribution for binned data using the standard $0.5\,\phi$ bins. In Panel (b), the interpolated upper and lower percentile confidence bounds for the binned data are shown as horizontal lines, and the associated 95% grain size confidence interval containing the true $D_{84}$ for the population is shown in grey.

compared with the grain size confidence interval associated with a sample size of 200, shown in Fig. 3b. We first develop a method to apply to samples comprising $n$ individual measurements of grain diameter, and then describe an approximation that can be applied to the more commonly encountered $0.5\phi$ binned cumulative grain size distributions.

### 2.1.2   Exact solution for a confidence interval

Suppose we wish to compute a confidence interval containing the population percentile, $D_P$, from our sample of 200 b-axis diameter measurements. The first step is to generate order statistics, $d_{(i)}$, by sorting the measurements into rank order from lowest to highest (such that $d_{(1)} \leq d_{(2)} \leq ... \leq d_{(n)}$). Figure 3a plots $d_{(i)}$ against the ratio $(i-1)/n$, which is a direct estimate of the proportion of the distribution that is finer than that grain size.

To define a confidence interval, we first specify the confidence level, usually expressed as $100\cdot(1-\alpha)\%$. For 95% confidence, $\alpha = 0.05$. Following Meeker et al. (2017), we then find lower and upper values of the order statistics ($d_{(l)}$ and $d_{(u)}$, respectively) that determine the percentile confidence interval, such that the coverage probability is as close as possible to $1-\alpha$, but no smaller. Note that, in our example of 100 coin tosses from the previous section, we made a calculation by setting $l = 40$ and $u = 60$, which gave us a coverage probability of 96%. Coverage probability is defined as:

$$P_c = \sum_{k=0}^{u-1} B_k(k,n,p) - \sum_{k=0}^{l-1} B_k(k,n,p) \tag{2}$$

where $B_k$ is the binomial probability distribution for $k$ successes in $n$ trials for probability $p$, defined in Eq. 1. The goal, then, is to find integer values $l$ and $u$ that satisfy the condition that $P_c \geq 1 - \alpha$, with the additional condition that $l$ and $u$ be approximately symmetric about the expected value of $k$ (i.e., $n \cdot p$). The lower grain size confidence bound for the estimate of

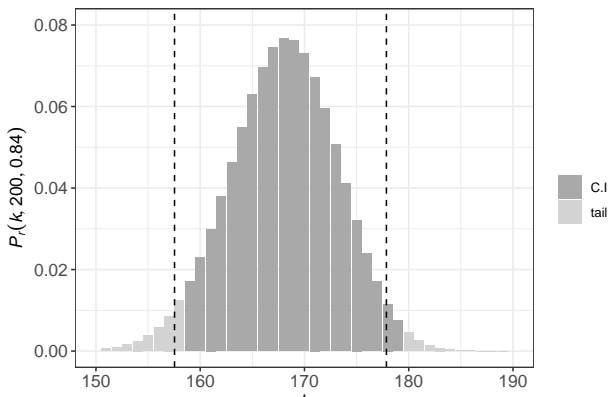

**Figure 4.** Binomial distribution values for $n = 200$ and $p = 0.84$, displaying the range of $k$ values included in the coverage probability. The dark grey bars indicate which order statistics are included in the 95% confidence interval, and light grey indicates the tails of the distribution that lie outside the interval. The vertical dashed lines indicate confidence limits computed by an approximate approach that places equal area under the two tails outside the confidence interval.

$D_P$ is then mapped to grain size measurement $d_{(l)}$ and upper bound is mapped to $d_{(u)}$. Obviously, this approach cannot be applied to the binned data usually collected in the field, but is intended for the the increasingly common automated, image-based techniques that retain individual grain size measurements.

We have created an R function (`QuantBD`) that determines the upper and lower confidence bounds, and returns the coverage probability, which is included in the `GSDtools` package. Our function is based on a script published online by W. Huber [1], which follows the approach described in Meeker et al. (2017). For $n = 200$, $p = 0.84$ and $\alpha = 0.05$ (i.e., 95% confidence level), $l = 159$ and $u = 180$, with a coverage probability (0.953) that is only slightly greater than the desired value of 0.95. This implies that the number of particles in a sample of 200 measurements that would be smaller than the true $D_{84}$ should range from 159 particles to 180 particles, 95% of the time. This in turn implies that the true $D_{84}$ could correspond to sample estimates ranging from the $80^{th}$ percentile (i.e., $159/200$) to the $90^{th}$ percentile (i.e., $180/200$). We can translate the percentile confidence bounds into corresponding grain size confidence bounds using our ranked grain size measurements: the lower bound of 159 corresponds to a measurement of 2.8 mm, and the upper bound corresponds to a measurement of 3.6 mm.

### 2.1.3 Approximate solution producing a symmetrical confidence interval

One disadvantage of the exact solution described above is that the areas under the tails of the binomial distribution differ (Fig. 4), such that the expected value is not located in the center of the confidence interval. Meeker et al. (2017) described a solution to this problem that uses interpolation to find lower or upper limits for one-sided intervals (i.e., confidence intervals pertaining to a one-tailed hypothesis test). This approach can be applied to find two-sided intervals by finding one-sided intervals, each with a confidence level of $1 - \alpha/2$, which result in a confidence interval that is symmetric about the expected value (see the

---

[1]https://stats.stackexchange.com/q/284970, last accessed on 19 September, 2018

dashed lines in Fig. 4). By interpolating between the integer values of $k$, we can find real numbers for which the binomial distribution has values of $\alpha/2$ and $1-\alpha/2$, which we refer to as $l_e$ and $u_e$. The corresponding grain sizes can be found by interpolating between measured diameters whose ranked order brackets the real numbers $l_e$ and $u_e$.

The values of $l_e$ and $u_e$ are indicated on Fig. 4 by dashed vertical lines. As can be seen, the values of $l$ and $u$ generated using the equal tail approximation are shifted to the left of those found by the exact approach, resulting in a symmetrical confidence interval. The corresponding grain sizes representing the confidence interval are 2.7 mm and 3.4 mm, which are similar to the exact solution presented above.

### 2.1.4 Approximate solution for binned data

We have adapted the approximate solution described above to allow estimation of confidence limits for binned data, which is accomplished by our R function called `WolmanCI` in the `GSDtools` package. Just as before, we use the equal area approximation of the binomial distribution to compute upper and lower limits ($l_e$ and $u_e$), but then we transform these ordinal values into percentiles by normalizing by the number of observations. Using our sample data, the ordinal confidence bounds $l_e = 157.03$ and $u_e = 177.36$ thus become the percentile confidence bounds $d_{79}$ and $d_{89}$, respectively.

Next, we simply interpolate from the binned cumulative frequency distribution to find the corresponding grain sizes that define the grain size confidence interval. Note that the linear interpolation is applied to $log_2(d)$, and that the interpolated values are then transformed to diameters in mm. This interpolation procedure is represented graphically in Fig. 3b; the horizontal lines represent the percentile confidence interval (defined by $l_e/n$ and $u_e/n$), while the grey box indicates the associated grain size confidence interval. Our binned sample data yield a grain size confidence interval for the $D_{84}$ that range from 2.8 mm to 3.5 mm.

The binomial probability approach uses the sample cumulative frequency distribution to calculate the grain size confidence interval. The need to use the sample distribution in this approach makes it difficult to predict the statistical power of sample size, $n$, prior to collecting the sample. However, the approach can be applied to any previously collected distribution, provided the number of observations used to generate the distribution is known.

## 3 Two-sample hypothesis tests

### 3.1 When individual grain diameters are available

Suppose we have two samples for which individual grain diameters have been measured (e.g., two sites, two operators, two sampling methods). The values in the two samples are denoted as $X_i$, (where $i$ ranges from 1 to $n_x$) and $Y_j$ ($j = 1$ to $n_y$) where $n_x$ and $n_y$ are the number of grains in each sample. In this case, one can use a resampling method (specifically the bootstrap) to develop a hypothesis test. A straightforward approach is based on the percentile bootstrap (Efron, 2016), and involves the following steps:

1. Take a random sample, $x$, which comprises a bootstrap sample of $n_x$ diameters, with replacement, from the set of values of $X_i$. The order statistics for the bootstrap sample are denoted as $x_{(k)}$, $k = 1$ to $n_x$.

2. Take a random sample, $y$, which comprises a bootstrap sample of $n_y$ diameters, with replacement, from the set of values of $Y_j$. The order statistics for the bootstrap sample are denoted as $y_{(l)}$, $l = 1$ to $n_y$.

3. Determine the desired percentile value from each bootstrap sample, $(d_P)_x$ and $(d_P)_y$, and compute the difference: $\Delta d_P = (d_P)_x - (d_P)_y$.

4. Repeat steps 1 to 3 $n_r$ times (e.g., $n_r = 1000$), each time storing the value of $\Delta d_P$.

5. Determine a confidence interval for $\Delta d_P$ by computing the quantiles corresponding to $\alpha/2$ and $1 - \alpha/2$, where $\alpha$ is the desired significance level for the test (e.g., $\alpha = 0.05$).

6. If the confidence interval determined in step 5 does not overlap 0, then one can reject the null hypothesis that the sampled populations have the same value of $D_P$.

This analysis is implemented with the function `CompareRAWs` in the `GSDtools` package. The required inputs are two vectors listing the measured $b$ axis diameters for each sample.

## 3.2 When only binned data are available and sample size is known

For situations in which only the cumulative frequency distribution is available, an approach similar to parametric bootstrapping can be applied, which employs the inverse transform approach (see Chapter 7 in Wicklin, 2013) to convert a set of random uniform numbers in the interval (0, 1) to a random sample of grain diameters by interpolating from the binned cumulative frequency distribution, similar to the procedure described above for determining confidence intervals for binned data. The approach involves the following steps:

1. Generate a set of $n_x$ uniform random numbers, $u_i$, $i = 1$ to $n_x$. Generate a bootstrap sample, $x$, by transforming these values of $u_i$ into a corresponding set of grain diameters $x_i$ by using the cumulative frequency distribution for the sample.

2. Generate a set of $n_y$ uniform random numbers, $u_j$, $j = 1$ to $n_y$. Generate a bootstrap sample, $y$, by transforming these values of $u_i$ into a corresponding set of grain diameters $y_j$ by using the cumulative frequency distribution for the second sample.

3. Determine the desired grain size percentile from each bootstrap sample, $(d_P)_x$ and $(d_P)_y$, and compute the difference: $\Delta d_P = (d_P)_x - (d_P)_y$.

4. Repeat steps 1 to 3 $n_r$ times (e.g., $n_r = 1000$), each time storing the value of $\Delta d_P$.

5. Determine a confidence interval for $\Delta d_P$ by computing the quantiles corresponding to $\alpha/2$ and $1 - \alpha/2$, where $\alpha$ is the desired significance level for the test (e.g., $\alpha = 0.05$).

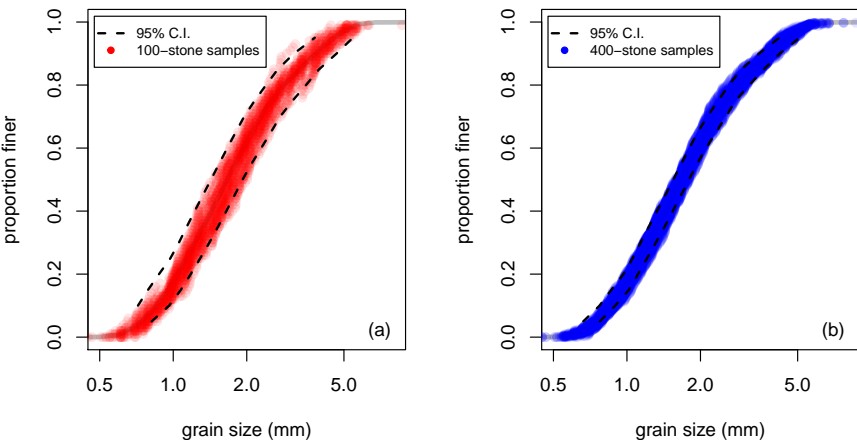

**Figure 5.** Effect of sample size on uncertainty. In Panel (a), 25 samples of 100 stones drawn from a known population are plotted, along with the 95% grain size confidence interval calculated for $D_5$ to $D_{95}$ using the binomial method. In Panel (b) samples of 400 stones are plotted, along with the predicted grain size confidence interval.

6. If the confidence interval determined in step 5 does not overlap 0, then one can reject the null hypothesis that the sampled populations have the same value of $D_P$.

This analysis is implemented in the `CompareCFDs` function. It requires that the user provide the cumulative frequency distribution for each sample (as a data frame), as well as the number of measurements upon which each distribution is based.

## 4   Confidence interval testing

We can test whether or not our approach successfully predicts the uncertainty associated with a given sample size using our known population of 3411 measurements from the lab. The effect of sample size on the spread of the data is demonstrated graphically in Fig. 5. In Fig. 5a, 25 random samples of 100 stones selected from the population are plotted, along with the 95% grain size confidence interval bracketing the true grain size population, calculated using our binomial approach and assuming a sample size of 100 stones. In Fig. 5b, random samples of 400 stones are plotted, along with the corresponding binomial confidence interval, calculated assuming a sample size of 400 stones. In both plots, the calculated confidence intervals are only plotted for the 5th percentile to the 95th percentile (a convention we use on all subsequent plots, as well), since few researchers ever make use of percentile estimates outside this range.

A comparison of the two plots shows that sample size (i.e. 100 vs. 400 stones) has a strong effect on variability of the sampled distributions. It is also clear that the variability of the sample data is well predicted by the binomial approach, since the sample data generally fall within the confidence interval for the population.

In order to test more formally the binomial approach, we generated 10,000 random samples (with replacement) from our population of 3411 observations, calculated sample percentiles ranging from the $d_5$ to the $d_{95}$ for each sample, and used the

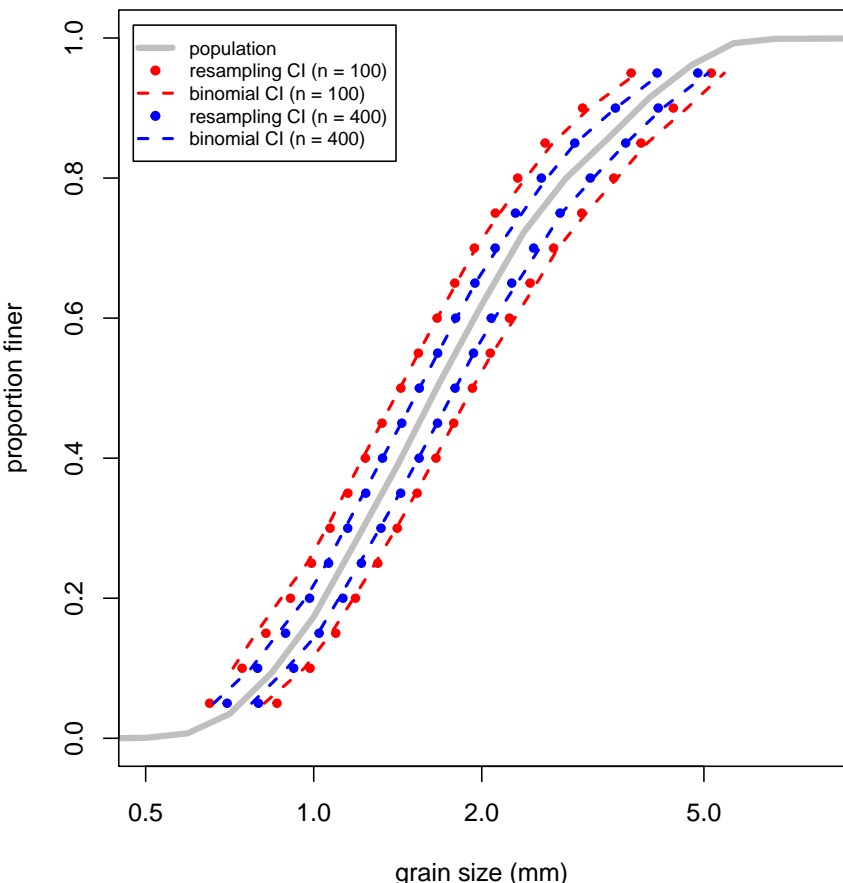

**Figure 6.** Comparing calculated resampling grain size confidence intervals to predicted intervals using the binomial approach. The grain size confidence intervals for samples of 100 stones are shown in red, and those for samples of 400 stones are shown in blue.

distribution of estimates to determine the grain size confidence interval. This resampling analysis was conducted twice; once for samples of 100 stones and then again for samples of 400 stones. This empirical approximation of the grain size confidence interval is the same technique used by Rice and Church (1996). The advantage of a resampling approach is that it replicates the act of sampling, and therefore does not introduce any additional assumptions or approximations. The accuracy of the

5   resampling approach is limited only by the number of samples collected, and the degree to which the individual estimates of a given percentile reproduce the distribution that would be produced by an infinite number of samples. The only draw back of this approach is that the results are only strictly applicable to the population to which the resampling analysis has been applied (Petrie and Diplas, 2000). While it is an ideal way to assess the effect of sample size on variability for a known population, resampling confidence intervals cannot be calculated for individual samples drawn from an unknown grain size population.

10   In Fig. 6, the resampling estimates of the 95% grain size confidence intervals for $D_5$ to $D_{95}$ based on samples of 100 stones are plotted as red circles, and those based on samples of 400 stones are plotted as blue circles. For comparison, the

confidence intervals predicted using our binomial approach are plotted using dashed lines. There is a close agreement between the resampling confidence intervals and the binomial confidence intervals, indicating that our implementation of binomial sampling theory captures the effects of sample size that we have numerically simulated using the resampling approach.

We have also calculated the statistics of a 1:1 linear fit between the upper and lower bounds of the confidence intervals predicted by binomial theory and those calculated using the resampling approach for sample sizes of 100 and 400. For a sample size of 100 stones, the 1:1 fit had a Nash Sutcliffe model efficiency ($NSE$) of 0.998, a root mean standard error ($RMSE$) of $0.0353\phi$ units, and a mean bias ($MB$) of $-0.0035\phi$ units. Since $NSE = 1$ indicates perfect model agreement (see Nash and Sutcliffe, 1970), and considering that $MB$ is small relative to the $RMSE$, these fit parameters indicate a good 1:1 agreement between the resampling estimates and binomial predictions of the upper and lower confidence interval bounds. The results for a sample size of 400 stones were essentially the same ($NSE = 0.999$, $RMSE = 0.0262\phi$, and $MB = 7e - 04\phi$).

In order to confirm that the size of the original population did not affect our comparison of the resampling and binomial confidence bounds estimates, we repeated the entire analysis using a simulated log-normal grain size distribution of 1,000,000 measurements. The graphical comparison of the binomial and resampling confidence intervals for the simulated distributions (not shown) was essentially the same as that shown in Fig. 6, and the 1:1 model fit was similar to the fits reported above ($NSE = 0.998$, $RMSE = 0.043\phi$, and $MB = -0.0013\phi$).

The close match between the grain size confidence intervals predicted using binomial theory and those estimated using the resampling analysis supports the validity of the proposed approach for computing confidence intervals.

## 5 Reassessing previous analyses

In order to demonstrate the importance of understanding the uncertainty associated with estimated grain size percentiles, we have reanalyzed the results of previous papers that have compared bed surface texture distributions, but which have not considered uncertainty associated with sampling variability. In some cases, these re-analyses confirm the authors' interpretations, and strengthen them by highlighting which parts of the distributions are different and which are similar, thus allowing for a more nuanced understanding. In others, they demonstrate that the observed differences do not appear to be statistically significant, and suggest that the interpretations and explanations of those differences are not supported by the authors' data. In either case, we believe that adding information about the grain size confidence intervals is a valuable step that should be included in every surface grain size distribution analysis.

The data published by Bunte et al. (2009) include pebble counts of about 400 stones for different channel units in two mountain streams (see Fig. 7). Adding the grain size confidence intervals to the distributions emphasizes the differences and similarities between the distributions. The plotted confidence intervals for the exposed channel bars do not overlap the confidence intervals for other two units: we can therefore conclude that bars are significantly finer than the pool units and the run/riffle units in both streams. Differences between the pools and the run/riffle units are less obvious.

Based on a visual inspection of the data in Fig. 7, it seems that clear differences in bed texture exist when comparing pool units and run/riffle units for the fraction of sediment less than about 22.6 mm; the distributions of sediment coarser than this are

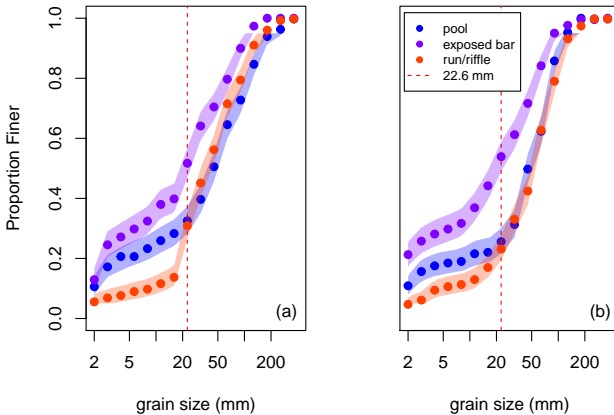

**Figure 7.** Comparing pebble counts from different channel units. Panel A presents data reported by Bunte et al. (2009) for Willow Creek. Panel B presents data for North St. Vrain Creek. Shaded polygons represent the 95% confidence intervals about the sample distribution, based on estimates ranging from $D_5$ to $D_{95}$. A dashed horizontal line indicate the approximate grain size at which pool distributions tend to diverge from run/riffle distributions for the two streams.

similar. Using the `CompareCFDs` function to statistically compare percentiles ranging from $D_5$ to $D_{95}$ (in increments of 5), we found that the differences between pools and run/riffle units in Willow Creek for percentiles greater than $D_{65}$ are significant for $\alpha = 0.05$, but not for $\alpha = 0.01$ (i.e., for a 99% confidence interval). For North St. Vrain Creek, there are significant differences between pool and run/riffle units at $\alpha = 0.05$ for percentiles finer than $D_{20}$, and for the $D_{80}$ and $D_{85}$, though none of the
differences for the coarser part of the distribution are significant for $\alpha = 0.01$.

The relative similarity of pool and run/riffle sediment textures for the coarser part of the distribution suggests that the most noticeable differences in bed surface texture are likely due to the deposition of finer bed-load sediment in pools on the waning limb of the previous flood hydrograph (as suggested by Beschta and Jackson, 1979; Lisle and Hilton, 1992, 1999), and that the bed surface texture of both units could be similar during flood events. From these plots we can conclude that the bed roughness
(which is typically indexed by the bed surface $D_{50}$ or by sediment coarser than that) is similar for the pool and run/riffle units, but that exposed bar surfaces in these two streams are systematically less rough. These kinds of inferences could have important implications for decisions about the spatial resolution of roughness estimates required to build 2D or 3D flow models; it is also possible to reach the same conclusions based on the original data plots in Bunte et al. (2009), but the addition of confidence bands supports the robustness of the inference.
A more fundamental motivation for plotting the binomial confidence bands is illustrated in Fig. 8, which compares the bed surface texture estimated by two different operators using the standard heel-to-toe technique to sample more than 400 stones from the same sedimentological unit. These data were published by Bunte and Abt (2001a) (see their Fig. 7). Based on their original representation of the two distributions (Fig. 8a), Bunte and Abt (2001a) concluded that

> "operators produced quite different sampling results ... operator B sampled more fine particles and fewer cobbles ... than operator A and produced thus a generally finer distribution."

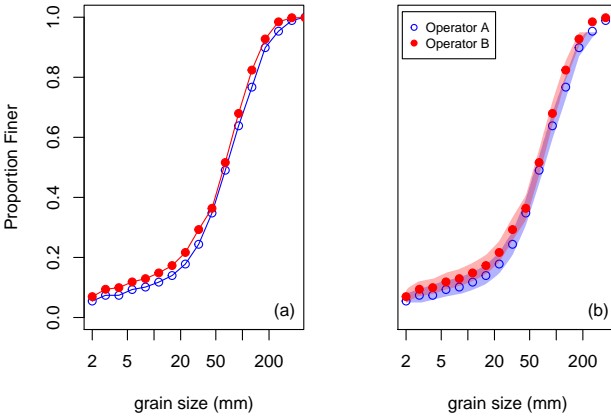

**Figure 8.** Comparing pebble counts of the same bed surface by different operators. The data plotted were published by Bunte and Abt (2001a). Panel A shows the traditional grain size distribution representation. Panel B uses the 95% grain size confidence intervals calculated for the pebble count to demonstrate that the two distributions are not statistically different.

However, once the grain size confidence intervals are plotted (Fig.8b), it is clear that the differences are not generally statistically significant. Using the `CompareCFDs` function to compare each percentile from $D_5$ to $D_{95}$, we found no statistically significant differences for any percentile at $\alpha = 0.01$; at $\alpha = 0.05$, only differences for the $D_{80}$, $D_{85}$ and $D_{95}$ are significant. When comparing multiple percentile values, the risk of incorrectly rejecting the null hypothesis for at least one of the comparisons will be higher than the significance level ($\alpha$) specified by the analyst. One approach to avoid this problem is to apply a Bonferroni correction in which $\alpha$ is replaced by $\alpha/m$, where m is the number of metrics being compared. Applying this correction, there is no statistically significant difference between the two samples for $\alpha = 0.05$. The value in considering sampling variability in the analysis is that it supports a more nuanced interpretation of differences in grain size distributions.

A similar analysis of the heel-to-toe sampling method and the sampling frame method advocated by Bunte and Abt (2001a) shows that the distributions produced by the two methods are not generally statistically different, either (Fig. 9). The `CompareCFDs` function only found significant differences for grain size percentiles coarser than $D_{70}$ for $\alpha = 0.05$, and between $D_{75}$ and $D_{90}$ for $\alpha = 0.01$. Once the Bonferroni correction is applied, none of the differences between the two samples would be considered significant at $\alpha = 0.05$. However, while the Bonferroni correction protects against making Type I errors (i.e., rejecting the null hypothesis when it is in fact false), it reduces the power of the test (i.e., the ability of the test to reject the null hypothesis when it is false) (e.g. Nakagawa, 2004). The topic of multiple comparisons in hypothesis testing is complex and beyond the scope of the current paper.

In both cases, the uncertainty associated with sampling variability appears to be greater than the difference between operators or between sampling methods, and thus one cannot claim these differences as evidence for statistically significant effects. It is likely the case that there are significant differences among operators or between sampling methods, but larger sample sizes would be required to reduce the magnitude of sampling variability in order to identify those differences.

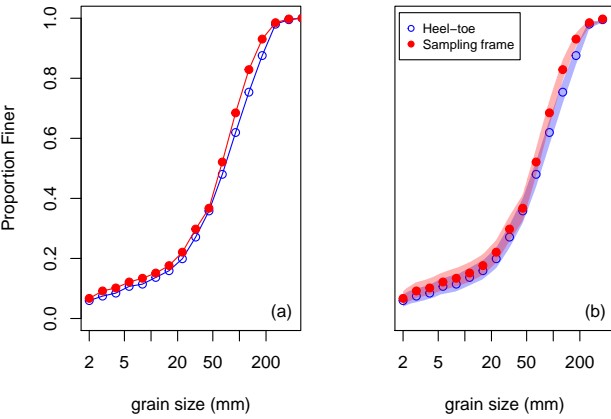

**Figure 9.** Comparing sampling methods for the same bed surface and operator. The data plotted were published by Bunte and Abt (2001a), and were collected by operator B. Panel A shows the traditional grain size distribution representation. Panel B uses the 95% grain size confidence intervals calculated for the pebble count to demonstrate that the two distributions do not appear to be statistically different.

Indeed, Hey and Thorne (1983) found that operator errors were difficult to detect for small sample sizes (wherein the sampling uncertainties were comparatively large), but became evident as sample size increased, so the issue at hand is not whether there are important differences between operators, but whether the differences in Fig. 8 are statistically significant. Interestingly, Hey and Thorne (1983) were able to detect operator differences at sample sizes of about 300 stones, whereas
Bunte and Abt (2001a) did not detect statistical differences for samples of about 400 stones, indicating either that Hey and Thorne (1983) had larger operator differences than did Bunte and Abt (2001a), or smaller sample uncertainties due to the nature of the sediment size distribution.

## 6   Determining sample size

As we demonstrated in the previous section, grain size confidence intervals can be constructed and plotted for virtually all
existing surface grain size distributions (provided that the number of stones that were measured is known, which is almost always the case), and future sampling efforts need not be modified in any way in order to take advantage of our method. While the primary purpose of our paper is to demonstrate the importance of calculating grain size confidence intervals when analyzing grain size data, our method can also be adapted to predict the sample size required to achieve a desired level of sampling precision, prior to collecting the sample.
While the percentile confidence interval for any percentile of interest can be calculated based on the sample size, $n$, and the desired confidence level, $\alpha$ (see Appendix B, for example), it cannot be mapped onto the grain size confidence interval before the cumulative distribution has been generated. This problem is well recognized, and has been approached in the past by making various assumptions about the distribution shape (Hey and Thorne, 1983; Church et al., 1987; Bunte and Abt, 2001a, b), or using empirical approximations (Marcus et al., 1995; Rice and Church, 1996; Green, 2003; Olsen et al., 2005), but in

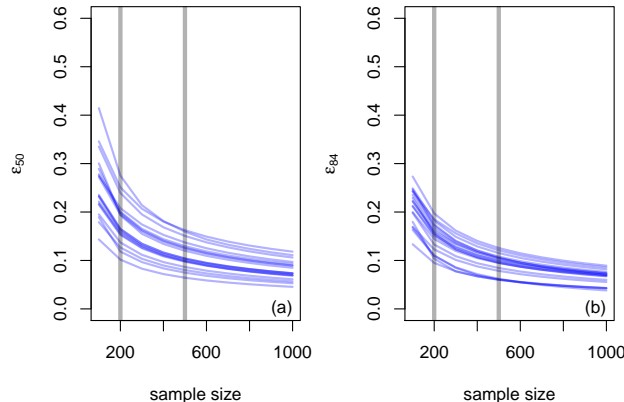

**Figure 10.** Estimated uncertainty for estimates of $D_{50}$ (Panel A) and $D_{84}$ (Panel B) are plotted against sample size. Curves were generated for bed surface samples collected by BGC Engineering and students from The University of British Columbia (unpublished data), and those published by Kondolf et al. (2003); Bunte and Abt (2001a); Bunte et al. (2009). Vertical lines highlight the range of uncertainties for sample sizes of 200 and 500 stones.

all cases it is still necessary to know something about the spread of the distribution – regardless of its assumed shape – in order to assess the implications of sample size for the precision of the resulting grain size estimates. It is perhaps the difficulty of predicting sample precision that has led to the persistent use of the standard 100-stone sample. Here we provide a simple means of determining the appropriate sample size; first we use existing data to calculate the uncertainty of estimates for $d_{50}$
and $d_{84}$; and then we use simulated log-normal grain size distributions to quantify the effect of the spread of the distribution on uncertainty.

### 6.1 Uncertainty based on field data

Here, we demonstrate the effect of sample size on uncertainty. We begin by calculating the uncertainty of estimates for $D_{50}$ and $D_{84}$ for all the surface samples used in this paper, for eight samples collected by BGC Engineering from gravel bed channels
in the Canadian Rocky Mountains, and for samples from two locations on Cheakamus River, British Columbia, collected by undergraduate students from the Department of Geography at The University of British Columbia. The number of stones actually measured to create these distributions is irrelevant, since it is the shape of the cumulative distribution that determines how the known percentile confidence interval maps onto the grain size confidence interval. Since these distributions come from a wide range of environments and have a range of distribution shapes, they are reasonable representation of the range grain size
confidence intervals that could be associated with a given percentile confidence interval.

Uncertainty ($\epsilon$) in the grain size estimate is calculated as follows:

$$\epsilon_P = 0.5 \left( \frac{d_{upper} - d_{lower}}{d_P} \right) \tag{3}$$

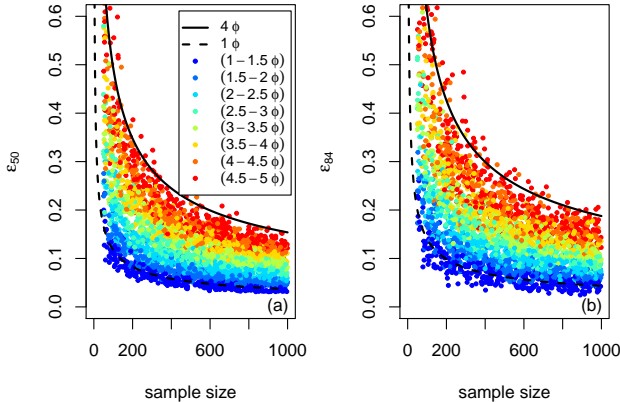

**Figure 11.** Estimated uncertainty for estimates of $D_{50}$ (Panel A) and $D_{84}$ (Panel B) are plotted against sample size for a simulated set of log normal surface distributions with a range of sorting indices. The markers are color-coded by $si_\phi$. The bounding curves for $si_\phi = 1$ and $si_\phi = 5.0$ are shown for reference, calculated using Eq. (6) and Eq. (8).

where $d_{upper}$ is the upper bound of the grain size confidence interval, $d_{lower}$ is the lower bound, and $d_P$ is the estimated grain size of the percentile of interest. As a result, $\epsilon_{50}$ represents the half-width of the grain size confidence interval about the median grain size (normalized by $d_{50}$), and $\epsilon_{84}$ represents half-width of the normalized grain size confidence interval for the $d_{84}$.

Fig. 10 presents the calculated values of $\epsilon_{50}$ and $\epsilon_{84}$ for various gravel bed surface samples, including those shown in Figs. (7), and (8). For a sample size of 100 stones, the uncertainties are relatively large, with a mean $\epsilon_{50}$ value of 0.25 and a mean
5   $\epsilon_{84}$ value of 0.21; for a sample of 200 stones, the mean $\epsilon_{50}$ value drops to 0.18, and the mean $\epsilon_{84}$ value drops to 0.15; and for $n = 500$, $\epsilon_{50} = 0.11$, and $\epsilon_{84} = 0.09$, on average. This analysis transforms the predictable, distribution-free contraction of the percentile confidence interval as sample size increases into the distribution-dependent contraction of the grain size confidence interval. Clearly there is a wide range of cumulative frequency distribution shapes in our data set, resulting in a large differences in $\epsilon_{50}$ and $\epsilon_{84}$ for the same sample size (and therefore the same percentile confidence interval).

## 10   6.2   Uncertainty for Log-normal distributions

In order to quantify the effect of distribution shape on the grain size confidence interval, we conducted a modelling analysis using simulated log-normal bed surface texture distributions that have a wide range of sorting index values. Here, sorting index ($si_\phi$) is defined by the following equation.

$$si_\phi = \phi_{84} - \phi_{16} \tag{4}$$

15   The term $\phi_{84}$ refers to the $84^{th}$ percentile grain size (in $\phi$ units), and $\phi_{16}$ refers the $16^{th}$ percentile. As a point of comparison, we estimated $si_\phi$ for the samples analyzed in the previous section. For those samples, the sorting index ranges from $1.5\phi$ to $5.6\phi$, with a median value of $2.5\phi$. The largest values of $si_\phi$ were associated with samples from channels on steep gravel bed

**Table 1.** Coefficient values for estimating uncertainty in $D_{50}$ and $D_{84}$ as a function of $si_\phi$ using Eqs. (6) and (8)

| $Coef.$ | $1.5\phi$ | $2.00\phi$ | $2.5\phi$ | $3.0\phi$ | $3.5\phi$ | $4.00\phi$ | $4.50\phi$ |
|---|---|---|---|---|---|---|---|
| A | 1.444 | 1.728 | 2.068 | 2.474 | 2.961 | 3.543 | 4.240 |
| B | 1.768 | 2.123 | 2.550 | 3.062 | 3.677 | 4.415 | 5.302 |

fans and on bar top surfaces, while samples characterizing the bed of typical gravel bed streams had values close to the median value.

We simulated 3000 log-normal grain size distributions with $D_{50}$ ranging from 22.6 mm to 90.5 mm, $n$ ranging from 50 to 1000 stones, and $si_\phi$ ranging from $1\phi$ to $5\phi$. For each simulated sample, we calculated uncertainty for $D_{50}$ and $D_{84}$ using Eq. 3. The calculated values of $\epsilon_{50}$ and $\epsilon_{84}$ are plotted in Fig. 11. Using the data shown in the figure, we fit least-squares regression to fit models of the form

$$\ln(\epsilon_P) = a \cdot n + b \cdot si_\phi + c \tag{5}$$

where $a$, $b$, and $c$ are the estimated coefficients. The empirical model predicting $\epsilon_{50}$ has an adjusted $R^2$ value of 0.95, with the variable $n$ explaining about 43% of the total variance, and $si_\phi$ explaining 51% of the variance. The model for $\epsilon_{84}$ has an adjusted $R^2$ value of 0.91 with the variables $n$ and $si_\phi$ explaining similar proportions of the total variance as they do in the $\epsilon_{50}$ model (41% and 50%, respectively).

10 After back-transforming from logarithms, the equation describing the $\epsilon_{50}$ can be expressed as:

$$\epsilon_{50} = A \cdot n^{-0.506} \tag{6}$$

where the coefficient $A$ is given by:

$$A = \exp(-0.171 + 0.359 si_\phi) \tag{7}$$

The equations for $\epsilon_{84}$ are:

$$\epsilon_{84} = B \cdot n^{-0.51} \tag{8}$$

where $B$ is given by:

$$B = \exp(0.021 + 0.366 si_\phi) \tag{9}$$

Table 1 provides values of $A$ and $B$ for a range of sorting indices.

## 7 Practical implications of uncertainty

20 The implications of uncertainty can be important in a range of practical applications. As an example, we translate grain size confidence intervals into confidence intervals for the critical discharge for significant morphologic change using data for Fishtrap Creek, a gravel bed stream in British Columbia that has been studied by the authors (Phillips and Eaton, 2009; Eaton

et al., 2010a, b). The estimated bed surface $D_{50}$ for Fishtrap Creek is about 55 mm, which we estimate becomes entrained at a shear stress of 40 Pa, corresponding to a discharge of about 2.5 m$^3$s$^{-1}$(Eaton et al., 2010b); the threshold discharge is based on visual observation of tracer stone movement, and corresponds to a critical dimensionless shear stress of approximately 0.045. If we assume that significant channel change can be expected when $D_{50}$ becomes fully mobile (which occurs at about twice the entrainment threshold, according to Wilcock and McArdell, 1993), then we would expect channel change to occur at a shear

stress of 80 Pa, which corresponds to a critical discharge of 8.3 m$^3$s$^{-1}$, based on the stage-discharge relations published by Phillips and Eaton (2009).

Since we used the standard technique of sampling 100 stones to estimate $D_{50}$ and since the sorting index of the bed surface is about 2.0$\phi$, we can assume that the uncertainty will be about $\pm17\%$, based on Eqs. 6 and 7, which in turn suggests that we can expect the actual surface $D_{50}$ to be as small as 46 mm or as large as 64 mm. This range of $D_{50}$ values translates to shear stresses

that produce full mobility that range from 67 Pa to 94 Pa. This in turn translates to critical discharge values for morphologic change ranging from 5.9 m$^3$s$^{-1}$to 11.2 m$^3$s$^{-1}$, which correspond to return periods of about 1.5 years and 7.4 years, based on the flood frequency analysis presented in Eaton et al. (2010b). Specifying a critical discharge for morphologic change that lies somewhere between a flood that occurs virtually every year and one that occurs about once a decade, on average, is of little practical use, and highlights the cost of relatively imprecise sampling techniques.

If we had taken a sample of 500 stones, we could assert that the true value of $D_{50}$ would likely fall between 51 mm and 59 mm, assuming an uncertainty of $\pm7\%$. The estimates of the critical discharge would range from 7.2 m$^3$s$^{-1}$ to 9.5 m$^3$s$^{-1}$, which in turn correspond to return periods of 2 years and 4.1 years, respectively. This constrains the problem more tightly, and is of much more practical use for managing the potential geohazards associated with channel change.

Operationally, it takes about 20 minutes for a crew of two or three people to sample 100 stones from a typical dry bar in a

gravel bed river, and a bit over an hour to sample 500 stones, so the effort required to sample the larger number of stones is often far from prohibitive. In less ideal conditions or when working alone, it may take upwards of 5 hours to collect a 500 stone sample, but as we have demonstrated, the uncertainty of the data increases quickly as sample size declines (see Figs. 10 and 11), which may make the extra effort worthwhile in many situations. Furthermore, computer-based analyses using photographs of the channel bed may be able to identify virtually all of the particles on the bed surface, and generate even larger samples.

The statistical advantages of the potential increase in sample size are obvious, and justify further concerted development of these computer-based methods, in our opinion.

## 8   Conclusions

Based on the statistical approach presented in this paper, we developed a suite of functions in the R language that can be used to first calculate the percentile confidence interval and then translate that into the grain size confidence interval for typical

pebble count samples (see the supplemental material for the source code). We also provide a spreadsheet which uses the normal approximation to the binomial distribution to estimate the grain size confidence interval. The approach presented in this paper uses binomial theory to calculate the percentile confidence interval for any percentile of interest (e.g. $P = 50$ or $P = 84$), and

then maps that confidence interval onto the cumulative grain size distribution based on pebble count data to estimate the grain size confidence interval. As a result, the approach requires only that the total number of stones used to generate the distribution is known in order to generate grain size distribution plots that indicate visually the precision of the sample distribution (e.g. Fig. 7). We have developed statistical approaches that can be used for samples in which individual grain sizes are known and for samples in which data are binned (e.g., into $\phi$ classes).

By estimating the grain size confidence intervals for each percentile in the distribution, the sample precision can be displayed graphically as a polygon surrounding the distribution estimates. When comparing two different distributions, this means of displaying grain size distribution data highlights which distributions appear statistically different, and which do not.

Our analysis of various samples collected in the field demonstrates that the grain size confidence interval depends on the shape of the distribution, with more widely graded sediments having wider grain size confidence intervals than narrowly graded ones. Our analysis also suggests that typical gravel bed river channels have a similar gradation, and that the typical uncertainty of the $D_{50}$ varies from $\pm 25\%$ for a sample size of 100 observations to about $\pm 11\%$ for 500 observations.

When designing a bed sampling program, it is useful to estimate the precision of the sampling strategy and to select the sample size accordingly; to do so, we must first assume something about the spread of the data (assuming a log-normal distribution), and then verify the uncertainty after collecting the samples. Simple equations for predicting uncertainty (as a percent of the estimate) are presented here to help workers select the appropriate sample size for the intended purpose of the data.

## Appendix A: Normal approximation

While it is difficult to determine the percentile confidence interval using Eq. 1 without using a scripting approach similar to the one we implement in the `GSDtools` package, we can approximate the percentile confidence interval analytically, and use the approximating equations in spreadsheet calculations. As Fripp and Diplas (1993) point out, the percentile of interest ($P$) can be approximated by a normally distributed variable with a standard deviation calculated as follows:

$$\sigma = 100 \frac{\sqrt{np(1-p)}}{n} \tag{A1}$$

The term $n$ refers to the number of stones being measured, and $p$ refers to the probability of a single stone being finer than the grain size for a percentile of interest, $D_P$ (recall from above that $p = P/100$, such that $p = 0.84$ for $D_{84}$). The standard deviation for $n = 100$ and $P = 84$ would be 3.7 . That means that the true $D_{84}$ would be expected to fall between sampled $d_{80.3}$ and $d_{87.7}$ for a sample of 100 observations approximately 68% of the time, and would fall outside that range 32% of the time.

More generally, we can use the normal approximation to calculate the percentile confidence interval for any chosen confidence level ($\alpha$). We simply need to find the appropriate value of the $z$ statistic for the chosen values of $\alpha$ and $n$, and calculate the percentile confidence interval using the following confidence bounds:

$$P_{upper} = P + \sigma z \tag{A2}$$

$$P_{lower} = P - \sigma z \tag{A3}$$

5    The use of a normal distribution to approximate the binomial distribution is generally assumed to be valid for $p$ values in the range $\frac{5}{n} \leq p \leq 1 - \frac{5}{n}$. For a sample size of 100 stones, the limits correspond to $5^{th}$ and $95^{th}$ percentiles of the distribution. However, some researchers have recommended the more stringent range of $\frac{20}{n} \leq p \leq 1 - \frac{20}{n}$ (e.g. Fripp and Diplas, 1993).

For ease of reference, Table A1 presents $\sigma$ values for $P$ ranging from 10 (i.e., the $D_{10}$) to 90 ($D_{90}$) and for $n$ ranging from 50 observations to 3200 observations. For $\alpha = 0.10$, $z = 1.64$; for a $\alpha = 0.05$, $z = 1.96$; and for $\alpha = 0.01$, $z = 2.58$. The table

5    can be used to estimate the approximate percentile confidence intervals for common values of $\alpha$, $P$ and $n$. However, the user will have to manually translate the percentile confidence intervals into grain size confidence intervals using the cumulative frequency distribution for their sample.

A spreadsheet (see supplemental material) implementing these calculations has also been developed. That spreadsheet maps the percentile confidence interval onto the user's grain size distribution sample in order to estimate the grain size confidence

10   interval.

**Table A1.** Percentile standard deviations for various sample sizes ($n$) and percentiles ($D_p$)

| $n$ | $D_{10}$ | $D_{16}$ | $D_{25}$ | $D_{50}$ | $D_{75}$ | $D_{84}$ | $D_{90}$ |
|---|---|---|---|---|---|---|---|
| 50 | 4.2 | 5.2 | 6.1 | 7.1 | 6.1 | 5.2 | 4.2 |
| 100 | 3.0 | 3.7 | 4.3 | 5.0 | 4.3 | 3.7 | 3.0 |
| 200 | 2.1 | 2.6 | 3.1 | 3.5 | 3.1 | 2.6 | 2.1 |
| 400 | 1.5 | 1.8 | 2.2 | 2.5 | 2.2 | 1.8 | 1.5 |
| 800 | 1.1 | 1.3 | 1.5 | 1.8 | 1.5 | 1.3 | 1.1 |
| 1600 | 0.8 | 0.9 | 1.1 | 1.2 | 1.1 | 0.9 | 0.7 |
| 3200 | 0.5 | 0.6 | 0.8 | 0.9 | 0.8 | 0.6 | 0.5 |

## Appendix B: Binomial distribution reference tables

This appendix presents reference tables for the percentile confidence interval calculations described above. The tables present calculations for a range of percentiles ($P$) and sample sizes ($n$). The calculations presented were made using the `GSDtools` package, hosted on Brett Eaton's GitHub page. It is freely accessible to download. You can also find a demonstration show-

5 ing how to install and use the package at `https://bceaton.github.io/GSDtools_demo_2019.nb.html`. The source code for the package can be found in the online data repository associated with this paper.

  These percentile confidence bounds do not depend on the characteristics of the grain size distribution, since they are determined by binomial sampling theory. Estimating the corresponding grain size confidence bounds requires the user to map the percentile confidence interval onto the grain size distribution in order to find the grain size confidence interval. The `GSDtools`

10 package will automatically estimate the grain size interval.

**Table B1.** Upper and lower percentile confidence interval bounds for $\alpha = 0.05$ (95% confidence level)

| | n = 100 | | n = 200 | | n = 300 | | n = 400 | | n = 500 | |
|---|---|---|---|---|---|---|---|---|---|---|
| $P$ | $P_{lower}$ | $P_{upper}$ | $P_{lower}$ | $P_{upper}$ | $P_{lower}$ | $P_{upper}$ | $P_{lower}$ | $P_{upper}$ | $P_{lower}$ | $P_{upper}$ |
| 10 | 4.0 | 15.8 | 5.8 | 14.1 | 6.6 | 13.3 | 7.0 | 12.9 | 7.3 | 12.6 |
| 15 | 7.8 | 21.8 | 10.0 | 19.9 | 10.9 | 19.0 | 11.5 | 18.5 | 11.8 | 18.1 |
| 20 | 12.0 | 27.6 | 14.3 | 25.4 | 15.4 | 24.5 | 16.0 | 23.9 | 16.4 | 23.5 |
| 25 | 16.2 | 33.2 | 18.9 | 30.9 | 20.0 | 29.8 | 20.7 | 29.2 | 21.2 | 28.7 |
| 30 | 20.7 | 38.7 | 23.5 | 36.2 | 24.7 | 35.1 | 25.4 | 34.4 | 25.9 | 34.0 |
| 35 | 25.3 | 44.0 | 28.2 | 41.4 | 29.5 | 40.3 | 30.2 | 39.6 | 30.7 | 39.1 |
| 40 | 30.0 | 49.2 | 33.0 | 46.6 | 34.3 | 45.4 | 35.1 | 44.7 | 35.6 | 44.2 |
| 45 | 34.8 | 54.3 | 37.9 | 51.7 | 39.2 | 50.5 | 40.0 | 49.8 | 40.5 | 49.3 |
| 50 | 39.7 | 59.3 | 42.8 | 56.7 | 44.2 | 55.5 | 45.0 | 54.8 | 45.5 | 54.3 |
| 55 | 44.7 | 64.2 | 47.8 | 61.6 | 49.2 | 60.5 | 50.0 | 59.7 | 50.5 | 59.3 |
| 60 | 49.8 | 69.0 | 52.9 | 66.5 | 54.3 | 65.3 | 55.0 | 64.7 | 55.6 | 64.2 |
| 65 | 55.0 | 73.7 | 58.1 | 71.3 | 59.4 | 70.2 | 60.2 | 69.5 | 60.7 | 69.1 |
| 70 | 60.3 | 78.3 | 63.3 | 76.0 | 64.6 | 75.0 | 65.3 | 74.3 | 65.8 | 73.9 |
| 75 | 65.8 | 82.8 | 68.6 | 80.6 | 69.8 | 79.7 | 70.6 | 79.1 | 71.1 | 78.6 |
| 80 | 71.4 | 87.0 | 74.1 | 85.2 | 75.2 | 84.3 | 75.9 | 83.7 | 76.3 | 83.4 |
| 85 | 77.2 | 91.2 | 79.6 | 89.5 | 80.7 | 88.8 | 81.3 | 88.3 | 81.7 | 88.0 |
| 90 | 83.2 | 95.0 | 85.4 | 93.7 | 86.3 | 93.1 | 86.8 | 92.7 | 87.2 | 92.5 |

**Table B2.** Upper and lower percentile confidence interval bounds for $\alpha = 0.10$ (90% confidence level)

| | n = 100 | | n = 200 | | n = 300 | | n = 400 | | n = 500 | |
|---|---|---|---|---|---|---|---|---|---|---|
| $P$ | $P_{lower}$ | $P_{upper}$ | $P_{lower}$ | $P_{upper}$ | $P_{lower}$ | $P_{upper}$ | $P_{lower}$ | $P_{upper}$ | $P_{lower}$ | $P_{upper}$ |
| 10 | 4.8 | 14.7 | 6.4 | 13.4 | 7.1 | 12.8 | 7.5 | 12.4 | 7.7 | 12.2 |
| 15 | 8.8 | 20.6 | 10.7 | 19.0 | 11.5 | 18.3 | 12.0 | 17.9 | 12.3 | 17.6 |
| 20 | 13.1 | 26.3 | 15.2 | 24.5 | 16.1 | 23.7 | 16.6 | 23.2 | 17.0 | 22.9 |
| 25 | 17.5 | 31.8 | 19.8 | 29.9 | 20.8 | 29.0 | 21.3 | 28.5 | 21.7 | 28.1 |
| 30 | 22.1 | 37.2 | 24.5 | 35.1 | 25.5 | 34.2 | 26.1 | 33.7 | 26.6 | 33.3 |
| 35 | 26.7 | 42.5 | 29.2 | 40.3 | 30.3 | 39.4 | 31.0 | 38.8 | 31.4 | 38.4 |
| 40 | 31.5 | 47.6 | 34.1 | 45.5 | 35.2 | 44.5 | 35.9 | 43.9 | 36.3 | 43.5 |
| 45 | 36.3 | 52.7 | 39.0 | 50.6 | 40.1 | 49.6 | 40.8 | 49.0 | 41.2 | 48.6 |
| 50 | 41.3 | 57.7 | 43.9 | 55.6 | 45.1 | 54.6 | 45.8 | 54.0 | 46.2 | 53.6 |
| 55 | 46.3 | 62.7 | 48.9 | 60.5 | 50.1 | 59.6 | 50.8 | 59.0 | 51.2 | 58.6 |
| 60 | 51.4 | 67.5 | 54.0 | 65.4 | 55.2 | 64.5 | 55.8 | 63.9 | 56.3 | 63.5 |
| 65 | 56.5 | 72.3 | 59.2 | 70.3 | 60.3 | 69.3 | 60.9 | 68.8 | 61.4 | 68.4 |
| 70 | 61.8 | 76.9 | 64.4 | 75.0 | 65.4 | 74.2 | 66.1 | 73.6 | 66.5 | 73.2 |
| 75 | 67.2 | 81.5 | 69.6 | 79.7 | 70.7 | 78.9 | 71.3 | 78.4 | 71.7 | 78.1 |
| 80 | 72.7 | 85.9 | 75.0 | 84.3 | 76.0 | 83.6 | 76.5 | 83.1 | 76.9 | 82.8 |
| 85 | 78.4 | 90.2 | 80.5 | 88.8 | 81.4 | 88.2 | 81.9 | 87.8 | 82.2 | 87.5 |
| 90 | 84.3 | 94.2 | 86.1 | 93.1 | 86.9 | 92.6 | 87.3 | 92.3 | 87.6 | 92.1 |

**Table B3.** Upper and lower percentile confidence interval bounds for $\alpha = 0.20$ (80% confidence level)

| | n = 100 | | n = 200 | | n = 300 | | n = 400 | | n = 500 | |
| $P$ | $P_{lower}$ | $P_{upper}$ | $P_{lower}$ | $P_{upper}$ | $P_{lower}$ | $P_{upper}$ | $P_{lower}$ | $P_{upper}$ | $P_{lower}$ | $P_{upper}$ |
|---|---|---|---|---|---|---|---|---|---|---|
| 10 | 5.7 | 13.5 | 7.1 | 12.5 | 7.6 | 12.1 | 8.0 | 11.8 | 8.2 | 11.6 |
| 15 | 10.0 | 19.2 | 11.5 | 18.0 | 12.2 | 17.5 | 12.6 | 17.2 | 12.9 | 17.0 |
| 20 | 14.4 | 24.7 | 16.1 | 23.4 | 16.9 | 22.8 | 17.3 | 22.5 | 17.6 | 22.2 |
| 25 | 19.0 | 30.1 | 20.8 | 28.7 | 21.6 | 28.1 | 22.1 | 27.7 | 22.4 | 27.4 |
| 30 | 23.6 | 35.4 | 25.6 | 33.9 | 26.5 | 33.2 | 26.9 | 32.8 | 27.3 | 32.5 |
| 35 | 28.4 | 40.7 | 30.4 | 39.1 | 31.3 | 38.4 | 31.8 | 37.9 | 32.2 | 37.6 |
| 40 | 33.2 | 45.8 | 35.3 | 44.2 | 36.2 | 43.5 | 36.7 | 43.0 | 37.1 | 42.7 |
| 45 | 38.1 | 50.9 | 40.2 | 49.3 | 41.2 | 48.5 | 41.7 | 48.1 | 42.0 | 47.8 |
| 50 | 43.1 | 55.9 | 45.2 | 54.3 | 46.1 | 53.5 | 46.7 | 53.1 | 47.0 | 52.8 |
| 55 | 48.1 | 60.9 | 50.2 | 59.3 | 51.1 | 58.5 | 51.7 | 58.1 | 52.0 | 57.8 |
| 60 | 53.2 | 65.8 | 55.3 | 64.2 | 56.2 | 63.5 | 56.7 | 63.0 | 57.1 | 62.7 |
| 65 | 58.3 | 70.6 | 60.4 | 69.1 | 61.3 | 68.4 | 61.8 | 67.9 | 62.2 | 67.6 |
| 70 | 63.6 | 75.4 | 65.6 | 73.9 | 66.4 | 73.2 | 66.9 | 72.8 | 67.3 | 72.5 |
| 75 | 68.9 | 80.0 | 70.8 | 78.7 | 71.6 | 78.0 | 72.1 | 77.6 | 72.4 | 77.4 |
| 80 | 74.3 | 84.6 | 76.1 | 83.4 | 76.8 | 82.8 | 77.3 | 82.4 | 77.6 | 82.2 |
| 85 | 79.8 | 89.0 | 81.5 | 88.0 | 82.2 | 87.5 | 82.6 | 87.1 | 82.8 | 86.9 |
| 90 | 85.5 | 93.3 | 87.0 | 92.4 | 87.6 | 92.0 | 87.9 | 91.8 | 88.2 | 91.6 |

**Table B4.** Upper and lower percentile confidence interval bounds for $\alpha = 0.33$ (67% confidence level)

| P | n = 100 | | n = 200 | | n = 300 | | n = 400 | | n = 500 | |
| --- | --- | --- | --- | --- | --- | --- | --- | --- | --- | --- |
|   | $P_{lower}$ | $P_{upper}$ | $P_{lower}$ | $P_{upper}$ | $P_{lower}$ | $P_{upper}$ | $P_{lower}$ | $P_{upper}$ | $P_{lower}$ | $P_{upper}$ |
| 10 | 6.5 | 12.4 | 7.7 | 11.8 | 8.1 | 11.5 | 8.4 | 11.3 | 8.6 | 11.2 |
| 15 | 11.0 | 18.0 | 12.3 | 17.2 | 12.8 | 16.8 | 13.1 | 16.6 | 13.3 | 16.5 |
| 20 | 15.6 | 23.4 | 17.0 | 22.5 | 17.6 | 22.1 | 17.9 | 21.8 | 18.2 | 21.6 |
| 25 | 20.3 | 28.7 | 21.8 | 27.7 | 22.4 | 27.3 | 22.8 | 27.0 | 23.0 | 26.8 |
| 30 | 25.0 | 34.0 | 26.6 | 32.9 | 27.3 | 32.4 | 27.6 | 32.1 | 27.9 | 31.9 |
| 35 | 29.8 | 39.2 | 31.5 | 38.0 | 32.1 | 37.5 | 32.5 | 37.2 | 32.8 | 37.0 |
| 40 | 34.7 | 44.3 | 36.4 | 43.1 | 37.1 | 42.6 | 37.5 | 42.3 | 37.8 | 42.0 |
| 45 | 39.6 | 49.4 | 41.3 | 48.2 | 42.0 | 47.6 | 42.4 | 47.3 | 42.7 | 47.1 |
| 50 | 44.6 | 54.4 | 46.3 | 53.2 | 47.0 | 52.6 | 47.4 | 52.3 | 47.7 | 52.1 |
| 55 | 49.6 | 59.4 | 51.3 | 58.2 | 52.0 | 57.6 | 52.5 | 57.3 | 52.7 | 57.1 |
| 60 | 54.7 | 64.3 | 56.4 | 63.1 | 57.1 | 62.6 | 57.5 | 62.3 | 57.8 | 62.0 |
| 65 | 59.8 | 69.2 | 61.5 | 68.0 | 62.1 | 67.5 | 62.6 | 67.2 | 62.8 | 67.0 |
| 70 | 65.0 | 74.0 | 66.6 | 72.9 | 67.3 | 72.4 | 67.6 | 72.1 | 67.9 | 71.9 |
| 75 | 70.3 | 78.7 | 71.8 | 77.7 | 72.4 | 77.3 | 72.8 | 77.0 | 73.0 | 76.8 |
| 80 | 75.6 | 83.4 | 77.0 | 82.5 | 77.6 | 82.1 | 77.9 | 81.8 | 78.2 | 81.6 |
| 85 | 81.0 | 88.0 | 82.3 | 87.2 | 82.8 | 86.8 | 83.1 | 86.6 | 83.3 | 86.5 |
| 90 | 86.6 | 92.5 | 87.7 | 91.8 | 88.1 | 91.5 | 88.4 | 91.3 | 88.6 | 91.2 |

*Code and data availability.* The source code for the R package, as well as the analysis code and the data used to create all of the figures in this paper are available online (doi:10.5281/zenodo.3234121). The R package used to estimate the confidence intervals for grain size distributions is also available online (doi:10.5281/zenodo.3229387).

*Author contributions.* B.C. Eaton drafted the manuscript, created the figures and tables, and wrote the code for the associated modelling and analysis in the manuscript; R.D. Moore developed the statistical basis for the approach, wrote the code to execute the error calculations, reviewed and edited the manuscript, and helped conceptualize the paper; and L.G. MacKenzie collected the laboratory data used in the paper,
5   tested the analysis methods presented in this paper, and reviewed and edited the manuscript.

*Competing interests.* The authors declare that they have no conflict of interest.

*Acknowledgements.* The field data used in this paper was provided by BGC Engineering, by undergraduate students in a field course run by one of the authors, and by an NSERC-funded research project being conducted by two of the authors. The paper benefited greatly from the input of one anonymous reviewer, and from Drs. K. Bunte and R. Hodge. We would also like to acknowledge the contributions of numerous
10   graduate students at UBC who tested our R package and commented on the revised version of the manuscript.

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
