# Peer review of "Percentile-based grain size distribution analysis tools (GSDtools) – estimating confidence limits and hypothesis tests for comparing two samples"

_Earth Surface Dynamics, 2019_

## Referee Comment (RC1) · Kristin Bunte (Referee) · 16 Mar 2019

General comments Eaton et al. (2019) present what seems to be a new way to compute confidence intervals around grain-size distributions that is based on the binomial approach. Encouraging the routine computation of confidence intervals around sampled grain-size distributions is a worthwhile undertaking and often a monitoring requirement for detecting change in rivers beds over time or space. The study by Eaton et al. sets out to provide such a tool. However, the authors do not succeed in making their tool easily accessible—in fact as presented, their approach remains a black box to most potential users. The manuscript does not provide more than general statistical background information and no step-by-step explanations are given on how a potential user could apply the authors' approach to his/her field data. The reader is not much

the wiser even after downloading the supplemental material which contains computer code but still no instructions on how to apply the computations. For a user whose basic work tool is spreadsheet computation, the study by Eaton et al. (2019) provides no help for computing confidence intervals.

Computation of confidence bands around grain-size distributions without assuming an underlying distribution type is not a new idea. Fripp and Diplas (1993) presented a binominal approach to compute the relation between sample size and error around individual percentiles. The study by Church and Rice (1996) applied a bootstrap approach to a large pebble count of 3500 particles and computed error bands around various percentiles of the grain size distribution. The grain-size distributions did not fit a particular distribution type, but the bootstrap confidence limits were reasonably close to those computed assuming an underlying skewed log-normal distribution. Petrie and Diplas (2002) cautioned that "... the binomial distribution considers only two possibilities for each particle sampled: (1) the particle is within a specific size class (e.g., smaller than a certain size) or (2) the particle is not within the specified size class. The binomial distribution is then inadequate to use for representing entire size distributions." To overcome this limitation and to compute confidence bands around the cumulative frequency distribution from a pebble count with data binned into size classes while considering distribution characteristics of the distribution, Petrie and Diplas (2000) developed a multinomial approach.

While the study presented by Eaton et al. (2019) is successful in raising awareness that the n=100 sample size is too low to attain reasonable accuracy for pebble counts in most gravel beds and that sample sizes of 400 or 500 particles are required to enable statistical evaluations about sameness or difference, the study does not succeed in presenting its computational approach in an easy to understand way. Providing computer code in R-language is not helpful for most users, hence the authors' computations cannot be repeated or applied by users who are not expert statisticians but are seeking to determine confidence limits around their sampled grain-size distributions.

The authors display the confidence bands that they drew with their binomial approach around grain-size distributions sampled in other studies (Kondolf, 1992; Bunte et al. 2009, Bunte and Abt, 2011) and go on to discuss whether the now-drawn confidence bands warrant the interpretations made in the original studies. In the final sections of the study, the authors show general relations between sampling error, as computed with their binomial approach, and sample size as well as distribution sorting.

Suggestions to the authors for improving their manuscript 1) Reference prior work and build on it Eaton et al. (2019) should discuss prior studies that likewise compute errors around percentiles without assuming an underlying distribution type and explain the improvements and advantages offered in the study presented. What reason is there for a user to select the authors' approach if the authors do not explain WHY their approach constitutes an improvement?

2) Provide explanations and instructions In order for readers to apply the binomial approach to their own data, the authors need to provide a step-by step explanation on how to use their approach rather than referring to a book on statistics, pointing to a website, and offering computer code in R-language. Offering a reader access to computer code is a courtesy, but not a substitute for a step-by step explanation, especially not for a very hands-on and applied topic of monitoring bed-material changes.

3) Comparison of results to those from prior work: How do percentile errors computed from the authors' binomial approach compare to percentile errors computed from other approaches? Apart from a similarity of sampling errors around the D50 and D84 that the authors computed from their binomial as well as a bootstrap approach for a symmetrical grain-size distribution (the authors' flume experiment), the authors do not show how their binomial approach to computing confidence bands relates to confidence bands computed from other approaches. The authors should apply their binomial approach together with the approaches suggested by Fripp and Diplas (1993), Petrie and Diplas (2000), and Rice and Church (1996) as well as simply to sample-size equations for an error around the mean to a few pebble-count distributions that differ in

their sorting and skewness (esp. the extent of a fine tail) and then assess differences and similarities between results.

4) Explain whether or how confidence intervals computed from the binomial approach are affected by sorting and skewness of a sampled grain-size distribution While the authors show that confidence bands increase in width with a distribution's sorting coefficient, the authors do not explain how exactly sorting (and skewness) of a sampled grain-size distribution (e.g., a tail of fines) flow into the computation of confidence intervals based on the binomial approach. The binomial approach introduced by Fripp and Diplas (1993) does not seem to involve sorting or skewness of the sampled distribution, suggesting that confidence intervals from a binomial approach are similar for all percentiles within a sampled grain-size distribution with a known sample size and number of size classes.

5) Have a user in mind and offer a procedure that is reasonably easy to be applied by the user The authors provide a study that is of interest to users who are involved in relations of sample size to error. However, the study is geared towards a statistically expert audience rather than the needs of non-expert potential users. If the authors' work is to be applied for monitoring purposes by staff from environmental agencies or consulting and by those whose main interest is not statistical but who need to apply such relations, then the authors need to provide detailed explanation and instruction. A spreadsheet implementation of their computations of a percentile error would be considerably more helpful than code in R-language.

6) Editing suggestions Figures provided by the authors are generally fine, but considering that the study discusses plotted details of whether or not confidence bands overlap, a larger figure size would be helpful. It would also be helpful to place the figures below their first mention in the text, not simply at the top of the page with a mention somewhere below on the page. With respect to writing style and typos (etc.), the manuscript is well written and clean.

Specific Comments p.2, l. 15: "…but the largest source of uncertainty in many cases is likely to be sampling variability, which is a function of sample size." How do the authors know that sampling variability (do they mean statistical uncertainty due to a poorly sorted channel bed?) rather than methodological differences (e.g., measurements of particle sizes, spatial heterogeneity, differences in the sampled channel width or leaving poorly accessible stream locations unsampled) is the most likely factor causing uncertainty? The comparative study by Bunte et al. (2009) showed that differences in sampling outcomes due to methodological variability can be huge.

p. 2, line 21: "… We then use this approach to demonstrate that the higher percentiles, such as D84, are subject to substantial uncertainty for typically used sample sizes, and that…" 1) Given this statement, it is odd then that the confidence bands drawn by the authors around the size-distributions from two streams sampled by Bunte et al. (2009) and another stream sampled by Bunte and Abt (2001) are all narrower for the D84 than for the D75 and the D95. 2) That statement is not backed by results from other studies: Rice and Church (1996) have shown for a very large pebble count that uncertainty was lowest for the D75 and D84 sizes, followed by the D50 and D95 sizes, and highest for the smallest percentiles. Green (1993) corroborated this finding; on average, the D73 could be determined with the least uncertainty. Similarly, Bunte and Abt (2001a) found in their field study that uncertainty was lowest for the D50 and the D75, slightly higher for the D84, D95 and D25, and percentiles lower than D25 were subject to the highest uncertainties.

p. 3, l. 3: "…since we preserve each measurement rather than grouping them into size classes, the data can be treated as a binomial experiment, …" Does that mean that the binominal computations is not applicable to field data binned in 0.5 phi units which results from measuring particle size using a 0.5-phi template?

In Eq. 1, Pr and p are not defined

p. 4, line 10-19: The description of the methodology is too vague. To allow a reader to

replicate the computations, authors need to provide step-by-step guidance. Reference to websites and other studies is not sufficient for a paper that would like to introduce a new approach to computing confidence bands.

p. 4, line 21: That statement comes out of the blue . . .what areas? What tails?? Fig. 2 does not provide much help either.

p. 5, line 1-5: Again, step-by-step instructions are needed to allow a reader to replicate the authors' approach.

p. 6, line 5-6 ". . .Based on the overlap in 5 confidence intervals for the eight samples, the distributions do not appear to be statistically different (see Fig. 3). . .. 1) Confidence bands plotted by the authors for their stream table sediment overlap for samples 2 and 3, but not for samples 1 and 4 (Fig. 3, panel A). 2) With respect to their multinomial approach, Petrie and Diplas (2000) stated that error bands are identical for all particle-size distributions as long as the value for alpha (e.g., 0.05) and the number of sampled size classes remain the same. For the authors' 8 samples from the stream table sediment surface, I assume that the same number of size classes were collected in each of the 8 samples and that the same alpha value was applied to all computed confidence bands. If the statement by Petrie and Diplas (2000) was true for the error bands conducted by the authors, then why do the error bands plotted in Fig. 3 differ between samples? 3) The authors use as basis for their analyses a sand-rich sand-gravel mixture with a D50 near 1.5 mm. The lengths of b-axes appear to have been determined to a precision of two decimals (e.g., 0.53 mm). It is difficult to imagine how a pebble count was performed and particle sizes were measured on sediment this small.

p. 7, Fig. 4: 1) While the box of box and whisker plots typically shows the quartiles, there is less standardization of what the whiskers represent. Please indicate what the whiskers in this plot represent. It can't be the overall spread because "outliers" are plotted as dots. Please define. 2) What parameter is plotted on the y-axis? Please clarify. 3) It would have been useful to show the 95% confidence limits, too.

p. 6, line 9-19. The authors state that they found a close match between the confidence bands computed from the binomial approach and a bootstrap approach (Fig. 4) for an unskewed grain-size distribution (i.e., their stream table sediment). The comparison plot by Petrie and Diplas (2000) for a pebble count from the Mamquam River shows that the confidence bands computed with the approach by Fripp and Diplas (1993) are between 0.02 and 0.06 phi-units higher than those from the bootstrap approach computed by Rice and Church (1996). Is the binomial approach by Fripp and Diplas (1993) similar or different to the authors' binomial approach? Does a binomial approach yield wider confidence bands than a bootstrap approach?

p. 7, lines 11-19: In the authors' reassessment of particle-size distributions from Kondolf (1997) and from Bunte et al. (2009), the authors need to clearly state to what percentage confidence the plotted confidence bands refer? I assume they are 95% confidence bands. Please clarify.

p. 8, Fig. 6: The study by Eaton et al. (2019) has drawn confidence limits around grain-size distributions from three Rocky Mountain gravel-bed streams sampled by Bunte et al. (2009) and Bunte and Abt (2001). 1) Based on visual examination of the error bands plotted in Fig. 6, I'd say that for Willow Creek, the error bands for riffles and pools are different except for the narrow range between 20 and 50 mm within which they cross. 2) The plotted confidence intervals for Willow Creek and the St. Vrain are jagged around the sampled distribution and seem to widen notably for the flatter sections of the cumulative size distribution but neck down for the steeper sections. The authors offer no explanation for this phenomenon.

p. 10, line 9-14: The authors write: "Our method for estimating uncertainty requires only the cumulative distribution and the number of measurements used to construct the distribution. Therefore, confidence intervals can be constructed and plotted for virtually all existing surface grain size distributions (provided that the number of stones that were measured is known, which is almost always the case),..."
If computation of the width of the confidence interval for any percentile of interest requires only knowledge of the sampled distribution and sample size n, and if the computation is conducted for each percentile individually, then how does the spread or sorting of the sampled distribution influence the computed confidence interval? Please CLARIFY!

p. 11, Fig. 9 and p. 12, Fig. 10: 1) The units in which the error is computed needs to be clearly stated. Somewhere down in the text the reader gets a hint that the error pertains to a percentage error in mm units. 2) The findings that percentile errors decrease with sample size and with the distribution sorting is in and of itself nothing new. What is new here is that the error is computed from the authors' binomial approach (assuming an underlying log-normal distribution for Fig. 10). To allow a reader to see whether there is a difference between errors computed from the authors' binomial approach and other approaches (e.g., Fripp and Diplas (1993) or simply errors around a mean), the computed relations between errors and n should be compared to errors computed with other approaches. 3) For comparison with other studies that compute percentile errors in terms of absolute +- error in phi-units it would be helpful if the error-n relations in Fig. 9 had a second y-axis with error in terms of the absolute +- error in phi-units. 4) It would be useful if the relation of error to n was also provided for the error around the D16.

p. 12, line 8: The authors state that for a given n and sorting, errors are largest for steep gravel fans and bar top surfaces and smaller for typical gravel beds with a sorting near 1. That is a useful comment. It would be even more useful to elaborate a little bit here on what kind of sorting values to expect for different morphological or sedimentary channel units and hence what a user needs to expect in terms of the error - sample size relation.

p. 14, line 12-13: I am afraid that the authors' time estimates refer to dry deposits of mainly mid-sized gravels. The time requirements for a 500-particle pebble count increases to about 5 hours when sampling in poorly wadeable conditions, in the presence

of abundant algae and large woody debris, under overhanging bushes, and with particles being next to irretrievable from the bed because they are tightly wedged within neighboring particles or small particles placed in tiny pockets between large clasts. The necessity for a large sample size remains, but users—and their funding agencies!—need to commit to realistic time requirement.

p. 14, Line 19: "a suit of functions in R language that can be used to estimate the uncertainty of any percentile in a cumulative grain-size distribution (see the supplemental material. . .)" I personally find "a suite of functions in R language" not useful, and I am not sure how many other potentially interested users have access to something in R-language. If the authors would like to provide helpful support to a user, then please provide a spreadsheet.

Typos etc. p. 2, l.5: The value should be 22.6 (=2ˆ0.5*16), not 22.7. p. 3 L. 5. . .compute the quantiles of the (Fig. 1). Something is off in that sentence. p. 4, Footnote: The access date is in the future.

---

## Referee Comment (RC2) · Anonymous Referee #2 · 21 Mar 2019

The submitted paper focuses on estimating uncertainties in measured grain size distributions using statistical analysis of grain size data from experiments, field measurements and synthetic data. I think that the authors make an important main point, which is that uncertainties in grain size distributions should be reported especially when used to assess grain size changes over time or in space. Although I am supportive of the overall goals, topics, and messages of this manuscript, I think that there are many details missing from the methods. This makes it difficult to evaluate how this calculation is actually applied, the assumptions involved, and finally how it compares to previously published studies on uncertainties in grain sizes. I suggest adding these details such that your paper can be understood by a broader audience.

Main comments Literature review: I would really like to see a more detailed review

of what previous studies have done to quantify uncertainties in the D50 and other percentiles of the grain size distributions. Do approaches without an assumed grain size distribution exist? If so, what is wrong with these approaches that motivates this current study? I'm a bit confused because in the introduction you state that there is no easy way to estimate the required sample size. In the abstract you also write that you propose a simple approach to estimate sample size, but this also relies on assuming a log-normal distribution as in previous studies highlighted on p 2 lines 8-9. What is the difference between your approach that assumes a log normal distribution to estimate sample size and other log normal approaches? It is not entirely clear to me in reading the introduction what is new in this study compared to previous approaches. A more in depth review of previous approaches and a statement of how this new approach is different would really help.

Calculations: I do not completely understand how some of the calculations are implemented and more details are needed in the main text. I have broken these comments into the main sections of the paper:

In section 2.1, how is equation (1) used? Please provide a stepwise explanation on how someone would perform these calculations and what information is needed. Right now it is somewhat difficult to understand how equation (2) is actually solved. Although I appreciate the inclusion of the R code that is part of this paper, a simple explanation of your detailed methodology is really needed in the main text to properly evaluate your methods. What are 'successes', please define. I am also somewhat confused about the definition of $p$, earlier you state it is the percentile of a distribution but on P 4 L6 is it called a probability.

In section 2.2, please also provide more details on this approach, one brief sentence on interpolation really does not make this calculation clear.

Section 3 and Figure 4 How many times did you create a sample with 100 grains to make these distributions in Figure 4? It seems like the results could really vary

with the number of 100 grain samples? Also, some explanation of the boxplots is needed to evaluate the results. What are the horizontal lines at the top and bottom ends of the distributions? This information is needed to validate that the two predictions actually provide similar results. Can you provide the actual numeric values of the 99% confidence interval bounds for the two methods in the figures to enable quantitative comparisons?

Specific comment denoted by page (P) and line numbers (L)

P 1 L 21-22 For facies mapping, my understanding of the Buffington approach is not that it is meant to be purely qualitative as implied here. They have visual classification of patches that are then verified by numerous pebble counts on the patches. So their approach likely provides a more accurate representation of the grain size distribution because they use many pebble counts in a single reach.

P3 L5 Missing word(s) here.

P4 L 12-16 Please state if this text is for a specific sample (e.g. the data shown in Figure 1), right now it seems to be written as if it applies to all grain size measurements but I don't think that is actually the case?

P4 L 15-16 Please explain what you mean by 19 times out of 20. I'm not clear why these exact numbers are chosen instead of a percent of trials. It is also not clear how this percent of trails was calculated or how the range of 159-180 was determined.

P4 L 21-23 Stating that the area under the tails differs is pretty vague. Do you mean tails of the distribution? How are the tails of the distribution defined? Please state why these different areas are problematic. Similarly, upper and lower limits of what exactly? What do you mean by a one-sided interval and how does this relate to your calculations? I can guess what you mean but the lack of language specificity here makes your text somewhat difficult to follow.

Figure 3 More details are needed as to how the grain size data were collected, through

a random sample or grid count? Were the samples in different locations on streamtable and using the same or different operators? It is a little difficult to see the confidence bounds in this figure to assess overlap of various distributions, not sure though how you can easily address this problem.

P 7 L 8 typo here

Figure 5. I appreciate this reanalysis but I don't think that you can say that the distributions are statistically similar or different without a similar confidence bound on the bulk sample data. Previous studies have demonstrated that bulk samples also have considerable uncertainty depending on the size of the actual bulk sample and the portion of the sample that is occupied by the largest grain sizes. So the bulk sample is also not free from uncertainties and this needs to be acknowledged.

P 8 L 3-5 The statement that fine sediment would be deposited preferentially in the pool rather than in the run/riffle during the waning limb of the preceding hydrograph needs some references to support it.

P 12 L 6-7 Please explain why you are assuming the standard deviation of the distribution is related to logD84-logD50.

P 12 L 10-12 I do not entirely why you are simulating log-normal samples with this given range of D50 values and SDlog values? How were these distributions simulated by defining D50 and SDlog beforehand? Figure 10 does not seem to be referenced or explained anywhere in the text.

P 13 L. 14-22 More details are needed as to how you estimated that this grain size is entrained at a certain shear stress and discharge. Did you use Shields equation? What critical Shields stress did you assume? How did you then translate this shear stress into a discharge beyond using a stage-discharge relation; did you have a measured channel bed slope and are you assuming stage is equivalent to the average flow depth in a reach? What is the basis of the assumption that D50 becomes fully mobile at twice

the shear stress needed to initiate D50 movement? Some rational and supporting references are needed to support this argument. I am also a little confused about this uncertainty in grain size because all of these sizes (46, 55, 64 mm) are essentially in the same half-phi bin. I may be mistaken but if you have binned your data into half phi intervals for this analysis, wouldn't you expect a similar, although likely smaller, level of uncertainty in the D50 anyway? This uncertainty would occur because you are determining the measured streambed D50 value (55 mm) by interpolation between the two percentiles straddling the 50th percentile value, and these two bounding percentiles correspond to grain size bins 45 and 64 mm. But you do not actually have any grain size resolution finer than half phi bin size. So when you calculate a median grain size of 55 mm, you are interpolating this grain size to a finer resolution than you actually have data. Doesn't this already seem to imply that your uncertainty in D50 might be somewhere within a half phi bin size when you only have binned data, depending of course on how the actual grain sizes are distributed within that half phi bin?

P 15 L 12-13 Although I certainly agree that having more than 100 sampled particles would be better for uncertainties in most studies, these time estimates assume a team of people performing pebble counts. Having conducted a very large number of pebble counts on my own, these can take much longer than 20 minutes. The time also really depends as to whether you are binning grain sizes or measuring individual b axes. Finally, setting up and finding grains on a grid also adds to the pebble count time, so I would argue that this 20 minute estimate is a minimum.

---

## Editor Comment (EC1) · Jens Turowski (Editor) · 21 Mar 2019

Dear authors,

we have now received two comprehensive reviews of the paper. both reviewers agree that this is a potentially important contribution, but they also highlight a number of shortcomings. The main criticisms are common in both reviews: 1) The new method for calculating uncertainties is insufficiently described and can, from the information currently available in the manuscript, not be reproduced. 2) It is unclear what aspects are new and what the improvements and benefits of the method are in comparison to existing methods. Within this context, both reviewers call for a more comprehensive literature review, and a comparison with existing methods. There are further queries

on the details of the statistics and data generation. Both of the points above seem to be essential to me if you want to make this paper a useful contribution for the community. When revising the paper, I ask you specifically to think from the perspective of a potential user. You could supply a flow chart or table that enumerates the steps that need to be taken to apply your method to a data set. It would also be good if you could supply a script as a supplement.

Please provide a detailed reply to the referees comments when you submit your revised paper.

I am looking forward to seeing you revisions, with best wishes, Jens Turowski

---

## Author Comment (AC1) · 24 Apr 2019

**1  General Comments**

The general comments from reviewer 1 are presented below, and are discussed at some length. We attempt to address all the key issues raised there, and to highlight how we are responding to those comments in our revisions. We thank the reviewer very much for such a careful and helpful review of our paper.

[Figure]

**1.1 Comment 1**

General comments Eaton et al. (2019) present what seems to be a new way to compute confidence intervals around grain-size distributions that is based on the binomial approach. Encouraging the routine computation of confidence intervals around sampled grain-size distributions is a worthwhile undertaking and often a monitoring requirement for detecting change in rivers beds over time or space. The study by Eaton et al.sets out to provide such a tool. However, the authors do not succeed in making their tool easily accessible: in fact as presented, their approach remains a black box to most potential users. The manuscript does not provide more than general statistical background information and no step-by-step explanations are given on how a potential user could apply the authors' approach to his/her field data. The reader is not much the wiser even after downloading the supplemental material which contains computer code but still no instructions on how to apply the computations. For a user whose basic work tool is spreadsheet computation, the study by Eaton et al. (2019) provides no help for computing confidence intervals.

**1.2 reply by authors**

This is very useful feedback for us. Our intention is indeed to provide a user-friendly tool that implements binomial statistical theory (which is the basis for the approximation presented by Fripp and Diplas, 1993, mentioned later by this reviewer) to calculate confidence bands about grain size distributions to prevent type 1 statistical errors (i.e. false positives, where differences between distributions is asserted where it is not statistically justified). The reviewer is incorrect in contending that we do "not provide more than general statistical background information", and that we do not provide "step-by-step explanations" on how to implement the analysis. The manuscript lays out the

**ESurfD**
precise statistical basis for the calculations we make using the different methodologies appropriate for different kinds of data (i.e. raw observations and binned data). However, it is clear that we have failed to communicate this appropriately for our target audience.

We are in the process of completely re-writing the introductory section of the paper to better explain

1. how the binomial distribution can be applied to both raw data comprising $n$ measurements of b-axis diameters and also to the typical binned data collected in the field; and

2. how the binomial theory can be used to generate confidence intervals about an estimate of a given grain size percentile.

The process is summarized using an example shown in a new figure we are integrating into the paper (Fig 1, below). In that figure, panel A presents the cumulative distribution of the population defined by 3411 individual measurements of b-axis diameters. Panel B shows a sample of 100 stones taken from that population and indicates the difference between the population median grain size and the sample median. Using the binomial equation, we calculate the uncertainty in terms of grain size percentile of the sample (a step that requires the implementation of the computer code that we provided, but which can be approximated using the normal distribution, as described by Fripp and Diplas, 1993). The percentile uncertainty is indicated in panel C using grey shading and arrows. Panel D indicates how this uncertainty in percentiles can be translated into a confidence interval for the sample that will contain the true population median grain size 95% of the time.

In addition, we are preparing an appendix to the paper that describes how to use the simpler normal approximation to the binomial distribution to calculate the confidence interval, and we are developing a spreadsheet implementation of that approach.

We have also created an appendix containing reference tables of the uncertainty (expressed as sample percentiles (corresponding to panel C of Fig 1) for a range of percentiles of interest (i.e. $D_{10}$, $D_{15}$, $D_{20}$ ... $D_{90}$), sample size ($n$), and acceptable confidence limit ($\alpha$).

Finally, we are developing a basic introduction to the R Package that demonstrates how to enter standard grain size data, set the confidence interval parameters (i.e., what $D_i$ values are of interest, and what $\alpha$ is acceptable), run the analysis to generate the confidence intervals, and then export the results as a text file that could be imported into any standard spreadsheet application. We appreciate that many people still use spreadsheets as their go-to analysis tool, and we are trying to accommodate those users. However, it is our experience that there is strong demand for the kind of R-based analysis tools that can be incorporated into scripting languages like R, so we have focused on the `bicalc` package as the primary means of distributing our tools.

**1.3 Comment 2**

Computation of confidence bands around grain-size distributions without assuming an underlying distribution type is not a new idea. Fripp and Diplas (1993) presented a binomial approach to compute the relation between sample size and error around individual percentiles. The study by Church and Rice (1996) applied a bootstrap approach to a large pebble count of 3500 particles and computed error bands around various percentiles of the grain size distribution. The grain-size distributions did not fit a particular distribution type, but the bootstrap confidence limits were reasonably close to those computed assuming an underlying skewed log-normal distribution. Petrie and Diplas (2002) cautioned that "...the binomial distribution considers only two possibilities for each particle sampled: (1) the particle is within a specific size class (e.g.,smaller than a certain size) or (2) the particle is

not within the specified size class. The binomial distribution is then inade-
quate to use for representing entire size distributions."To overcome this lim-
itation and to compute confidence bands around the cumulative frequency
distribution from a pebble count with data binned into size classes while
considering distribution characteristics of the distribution, Petrie and Diplas
(2000) developed a multinomial approach.

1.4   reply by authors

This is also very useful information for us, and we have read the papers with interest.
The work by Diplas and colleagues is particularly relevant and strengthens our paper.
There are some important differences, but the analysis by Fripp and Diplas (1993) can
be used as a jumping off point for our analysis. They use a normal approximation to
the binomial equation and apply it to binned data like that typically collected in the field.
We are re-writing our manuscript to use that paper as the basis from which we start,
we describe that approach in the appendix we are writing, and we will implement a
version of it in a spreadsheet that we are developing to accompany this paper.

The paper by Rice and Church (1996) was the inspiration for the bootstrap re-sampling
that we presented in our original paper (i.e. it is the boxplot in our original figure 4).
However, we have clearly not done justice to the analysis presented therein, so we
are expanding that section. Since our intention initially was to present a tool using
relatively standard statistical approaches for generating confidence bands around dis-
tributions, we focused on validating the approach rather than comparing it to previous
attempts. To be clear, Rice and Church (1996) did not present a *method* for esti-
mating uncertainties about different grain size percentiles, they presented *estimates*
of those uncertainties based on resampling measurements from a population of over
3000 b-axis diameter values (as described clearly by Petrie and Diplas, 2000). Those
estimates can only strictly be applied to the population of stones that they collected,

and only approximately to populations with similar characteristics. Our analysis in section 5.1 and Figure 9 show that many, but not all, gravel beds are similar to their data from Mamquam River. To apply their method to another river would require that a sample large enough to nearly perfectly define the population be taken (i.e. about 3500 stones), and then resampled with different sample sizes to estimate the uncertainty associated with the given sample size. Since the binomial theory approach replicates their sampling estimates (see Petrie and Diplas, 2000, to confirm this), then it is far more efficient to implement the binomial theory than to undertake a laborious empirical estimation. In our revised paper, we will make these issues clear.

The work by Petrie and Diplas with multinomial theory is primarily focused on determining the sample size required for a given level of accuracy for estimating the shape and relative position of the cumulative grain size distribution, using binned data. Our approach and intent is different: we develop our statistical theory using individual measurements of b-axis diameters, and we develop confidence bounds to be plotted when comparing distributions to avoid type 1 and 2 statistical errors. In this context, the binomial approach is most appropriate – a point made clearly by our successful validation of our approach against the statistics of repeated resampling of a known population (i.e. our Figure 9). Given the data that we use, it is not true to say "...the binomial distribution considers only two possibilities for each particle sampled: (1) the particle is within a specific size class (or (2) the particle is not within the specified size class." Our implementation of binomial theory is based on the interpretation that a measured stone is either (a) greater than a percentile of interest for the population, or (b) less than or equal to the percentile of interest, with no reference to or limitation imposed by having binned data. In this context, the estimation of $j$ percentiles involves the execution of $j$ independent binomial experiments with assumed probabilities corresponding to the percentile of interest. To test the difference between our approach and the traditional binned data, we will use the interpolation scheme described in our paper to directly compare the distributions based on all measurements, and the binned data (this was done implicitly in Fig 1 of our original paper, but we are much more explicit about this
in our revisions).

While current practice in the field is still to collect binned data, the automated techniques for grain size analysis that are standard practice in most experimental laboratories, and which are being increasingly deployed in the field promise to deliver much more data than can be collected manually, and will obviate the need for binned data. Our methodology is best leveraged in that context, using the automated data analysis approach possible using languages like R and Python. Therefore, our differentiation between binned data and the underlying b-axis diameter measurements is not simply a technical one, it is based on our perceptions of the future data types that will be commonly used.

**1.5 Comment 3**

While the study presented by Eaton et al. (2019) is successful in raising awareness that the n=100 sample size is too low to attain reasonable accuracy for pebble counts in most gravel beds and that sample sizes of 400 or 500 particles are required to enable statistical evaluations about sameness or difference, the study does not succeed in presenting its computational approach in an easy to understand way. Providing computer code in R-language is not helpful for most users, hence the authors' computations cannot be repeated or applied by users who are not expert statisticians but are seeking to determine confidence limits around their sampled grain-size distributions.The authors display the confidence bands that they drew with their binomial approach around grain-size distributions sampled in other studies (Kondolf, 1992; Bunte et al.2009, Bunte and Abt, 2011) and go on to discuss whether the now-drawn confidence bands warrant the interpretations made in the original studies. In the final sections of the study, the authors show general relations between sampling error, as computed with

their binomial approach, and sample size as well as distribution sorting.

**1.6   reply by authors**

We are very grateful for the feedback about the relative difficulty in understanding our approach, and about the need for addition means of implementing our tools for estimating the confidence bands. We are trying to respond to the first point by re-writing the section of the paper presenting the method, and to the second by developing reference materials in two appendices, a "how-to" guide for using the R code, and a spreadsheet implementing the normal approximation to our solution, as described by Fripp and Diplas (1993).

**2   Recommendations for improving the paper**

The reviewer made several helpful suggestions for improving the paper, listed below:

> Reference prior work and build on it Eaton et al. (2019) should discuss prior studies that likewise compute errors around percentiles without assuming an underlying distribution type and explain the improvements and advantages offered in the study presented. What reason is therefor a user to select the authors' approach if the authors do not explain WHY their approach constitutes an improvement?

As described (in part) above, we are improving the links between our paper and the previous work. We are also re-iterating in the paper that our main purpose is to produce a user-friendly introduction to the basic method for estimating confidence bounds using binomial theory (as approximated by Fripp and Diplas, 1993; and validated empirically by Rice and Church's empirical analysis, as well as our own parallel empirical

analysis of our own relatively large population of b-axis measurements). We will point out that our approach is statistcally conventional, has precedents in the literature, and is consistent with empriical analyses. We will also more strongly articulate that the key message of the paper is that all grain size curves ought to be plotted with confidence intervals, particularly when two distributions are being compared. The method by which the confidence interval is calculated is less important than the fact that it is calculated at all. Current practice seems to be to ignore all statistical uncertainty (despite the precedents in the literature); we hope this paper will make it easy for researchers to actually conduct the statistical analysis, and include it in their analysese.

> Provide explanations and instructions In order for readers to apply the binomial approach to their own data, the authors need to provide a step-by step explanation on how to use their approach rather than referring to a book on statistics, pointing to a website, and offering computer code in R-language. Offering a reader access to computer code is a courtesy, but not a substitute for a step-by step explanation, especially not for a very hands-on and applied topic of monitoring bed-material changes.

With this particular comment in mind, we have re-written the manuscript and generated various reference materials. We are currently "test-driving" the new material with graduate student who do not have extensive statistical backgrounds.

> Comparison of results to those from prior work: How do percentile errors computed from the authors' binomial approach compare to percentile errors computed from other approaches? Apart from a similarity of sampling errors around the D50 and D84that the authors computed from their binomial as well as a bootstrap approach for asymmetrical grain-size distribution (the authors' flume experiment), the authors do not show how their binomial approach to computing confidence bands relates to confidence

bands computed from other approaches. The authors should apply their binomial approach together with the approaches suggested by Fripp and Diplas (1993),Petrie and Diplas (2000), and Rice and Church (1996) as well as simply to sample-size equations for an error around the mean to a few pebble-count distributions that differ in their sorting and skewness (esp. the extent of a fine tail) and then assess difference sand similarities between results.

In our revised paper, we make the links to the cited literature clear, and we make it more clear that we did in fact replicate the approach described by Rice and Church, and then compare it to the binomial methodology we describe.

Explain whether or how confidence intervals computed from the binomial approach are affected by sorting and skewness of a sampled grain-size distribution While the authors show that confidence bands increase in width with a distribution's sorting co-efficient, the authors do not explain how exactly sorting (and skewness) of a sampled grain-size distribution (e.g., a tail of fines) flow into the computation of confidence intervals based on the binomial approach. The binomial approach introduced by Fripp and Diplas (1993) does not seem to involve sorting or skewness of the sampled distribution, suggesting that confidence intervals from a binomial approach are similar for all percentiles within a sampled grain-size distribution with a known sample size and number of size classes.

The revised description of our methodology and the new figures we are developing (e.g. Fig. 1, attached), will address this point.

Have a user in mind and offer a procedure that is reasonably easy to be applied by the user The authors provide a study that is of interest to users

who are involved in relations of sample size to error. However, the study is geared towards a statistically expert audience rather than the needs of non-expert potential users. If the authors'work is to be applied for monitoring purposes by staff from environmental agencies or consulting and by those whose main interest is not statistical but who need to apply such relations, then the authors need to provide detailed explanation and instruction.A spreadsheet implementation of their computations of a percentile error would be considerably more helpful than code in R-language.

We are developing resources that address this point, and we are particularly thankfull for this feedback, since our main purpose is to make it easy for people to use our approach. Sometimes we forget that distributing an R package is a great solution for only a sub-set of the community we hope to reach.

Editing suggestions Figures provided by the authors are generally fine, but considering that the study discusses plotted details of whether or not confidence bands overlap,a larger figure size would be helpful. It would also be helpful to place the figures below their first mention in the text, not simply at the top of the page with a mention some-where below on the page. With respect to writing style and typos (etc.), the manuscript is well written and clean

We are re-working our figures, and will leave it to the editorial staff to properly place the figures in the final version of the manuscript.
**3 Specific comments**

The reviewer also provides a list of specific comments that will improve the paper. We are currently working to integrate those specific comments.

**Fig. 1.** Example of binomial equation used to generate confidence bounds.

---

## Author Comment (AC2) · 24 Apr 2019

**1 General Comments**

The comments provided by Reviewer 2 are presented below, along with our responses. Several of key points raised by Reviewer 1 (and addressed in detail in our responses to R1's comments, which are already posted) come up again. This highlights the importance of those points, and we are very grateful for the feedback.

**1.1 Comment 1**

The submitted paper focuses on estimating uncertainties in measured grain size distributions using statistical analysis of grain size data from experiments, field measurements and synthetic data. I think that the authors make an important main point, which is that uncertainties in grain size distributions should be reported especially when used to assess grain size changes over time or in space. Although I am supportive of the overall goals, topics, and messages of this manuscript, I think that there are many details missing from the methods. This makes it difficult to evaluate how this calculation is actually applied, the assumptions involved, and finally how it compares to previously published studies on uncertainties in grain sizes. I suggest adding these details such that your paper can be understood by a broader audience.

1.2 reply by authors

We are quite gratified to see that Reviewer 2 is supportive of including error analysis into the treatment of grain size data. That is the primary motivation for this paper, and for the development of the bicalc package. The general criticisms raised are very similar to the comments we had from Reviewer 1, and we are making major changes to help address them. This includes a comprehensive re-write of the statistical basis section to better describe how our approach actually works, expansion of the results section to clarify the links to previous work, the development of reference appendices providing supporting information, the creation of a spreadsheet that implements the normal approximation to our technique (as described by Fripp and Diplas, 1993) to estimating confidence limits, and the development of a 'how-to' guide for using the R package. We appreciate all of the suggestions that are made in this review, and we are confident that the revised version will reach a broader audience.

**ESurfD**
**1.3 Comment 2**

I would really like to see a more detailed review of what previous studies have done to quantify uncertainties in the D50 and other percentiles of the grain size distributions. Do approaches without an assumed grain size distribution exist? If so, what is wrong with these approaches that motivates this current study? I'm a bit confused because in the introduction you state that there is no easy way to estimate the required sample size. In the abstract you also write that you propose a simple approach to estimate sample size, but this also relies on assuming a log-normal distribution as in previous studies highlighted on p 2 lines 8-9. What is the difference between your approach that assumes a log normal distribution to estimate sample size and other log normal approaches? It is not entirely clear to me in reading the introduction what is new in this study compared to previous approaches. A more in depth review of previous approaches and a statement of how this new approach is different would really help.

1.4 reply by authors

As described in our reply to Reviewer 1, we are extending our discussion of previous approaches by re-writing the paper to leverage the previous work by Diplas and colleagues as the starting point, and we describe in more detail how we replicated the bootstrap approach of Rice and Church to estimate the uncertainty of samples with various sizes drawn from our population of 3411 b-axis measurements. Basically, we believe that our approach is entirely consistent with that proposed by Fripp and Diplas (1993), and replicates the empirical results presented by Rice and Church (1996), but with a simple set of calculations rather than the laborious collection of 1000's of b-axis measurements. The main issue that we try to address in this paper is not that previous methods are flawed, but rather that we as a community have failed to use those

**ESurfD**
approaches to quantify sampling uncertainty (despite the precedents in the literature). As a result, there are published results that are clearly not statistically defensible, and it is our impression that many people continue to collect relatively small samples with limited appreciation of what that means in terms of uncertainty.

In our revisions, we will also emphasize that we think are the main contributions of this paper, which are:

- to describe clearly how surface sampling can be described as a binomial experiment, analogous to a traditional coin toss experiment (a point that the lead author of the paper failed to fully appreciate until we started working on this paper);
- to present a simple set of tools based on binomial theory with which anybody can easily calculate the confidence interval about any sample estimate of a grain size percentile that will contain the population percentile size for a given confidence level, α (this now includes a better description of how the method actually works, reference tables, a guide to using the R package, and a spreadsheet that implements a version of Fripp and Diplas's (1993) normal approximation to the binomial solution);
- to demonstrate the importance of considering uncertainty when comparing samples of the bed surface, or when making calculations based on those samples; and finally
- to make some assumptions about distribution shape so that we can provide some general guidance on the sample size required to reach a desired level of sampling precision.

This last point involves making assumptions about the underlying distribution (i.e. we assume a log normal grain size curve), but that is simply to generate synthetic data
with which to model the effect of sample size and the spread of the distribution on the precision of a percentile estimate. We will make it clear that any distribution form could have been used, but that we chose a log-normal distribution because (1) is the simplest to describe (i.e. it can be described by a mean and standard deviation), (2) it has been used previously by others, and (3) many gravel beds are approximately log-normal. We are attempting to more clearly emphasize our central message in our revisions (i.e. that we need to calculate confidence intervals, and that we can do so easily with binomial theory), and de-emphasize the point about sample size (since the actual precision of a sample in terms of grain size can only be determined once the sample has been collected).

**1.5 Comment 4**

The reviewer made several comments about our calculations that we would like to address:

In section 2.1, how is equation (1) used? Please provide a step wise explanation nohow someone would perform these calculations and what information is needed. Right now it is somewhat difficult to understand how equation (2) is actually solved. Although I appreciate the inclusion of the R code that is part of this paper, a simple explanation of your detailed methodology is really needed in the main text to properly evaluate your methods. What are 'successes', please define. I am also somewhat confused about the definition of p, earlier you state it is the percentile of a distribution but on P 4 L6 is it called a probability.

We have completely re-written the statistical basis, including an overview section that walks the user through the idea of a binomial experiment, the probabilities of a particular outcome (and the relation of those probabilities to the grain size percentiles for the

**ESurfD**
population being sampled), and the relation between confidence interval of the sample percentiles and the confidence interval bounding an estimate of a grain size percentile.

In section 2.2, please also provide more details on this approach, one brief sentence on interpolation really does not make this calculation clear.

We are re-writing the entire section to improve clarity.

Section 3 and Figure 4 How many times did you create a sample with 100 grains to make these distributions in Figure 4? It seems like the results could really vary with the number of 100 grain samples? Also, some explanation of the boxplots is needed to evaluate the results. What are the horizontal lines at the top and bottom ends of the distributions? This information is needed to validate that the two predictions actually provide similar results. Can you provide the actual numeric values of the 99% confidence interval bounds for the two methods in the figures to enable quantitative comparisons?

We are expanding and re-writing the entire section to improve it. This is the section where we repeat the kind of boot strap error estimates presented by Rice and Church (1996). We ended up taking 5000 samples from the population to ensure that the distributions of estimates stabilized.

**2 Specific comments**

The reviewer also provides a list of specific comments that will improve the paper. We are currently working to integrate those specific comments.

**ESurfD**

---

## Author Response (AR1)

**Estimating confidence intervals for gravel bed surface grain size distributions: Reply to reviewer comments**

Brett C. Eaton[1], R. Dan Moore[1], and Lucy G. MacKenzie[1]

[1]Geography, The University of British Columbia, 1984 West Mall, Vancouver, BC, Canada

May 28, 2019

**1 Reviewer 1: General Comments**

The general comments from reviewer 1 are presented below, and are discussed at some length. We attempt to address all the key issues raised there, and to highlight how we are responding to those comments in our revisions. We thank the reviewer very much for such a careful and helpful review of our paper.

**1.1 Comment 1**

General comments Eaton et al. (2019) present what seems to be a new way to compute confidence intervals around grain-size distributions that is based on the binomial approach. Encouraging the routine computation of confidence intervals around sampled grain-size distributions is a worthwhile undertaking and often a monitoring requirement for detecting change in rivers beds over time or space. The study by Eaton et al.sets out to provide such a tool. However, the authors do not succeed in making their tool easily accessible: in fact as presented, their approach remains a black box to most potential users. The manuscript does not provide more than general statistical background information and no step-by-step explanations are given on how a potential user could apply the authors. approach to his/her field data. The reader is not much the wiser even after downloading the supplemental material which contains computer code but still no instructions on how to apply the computations. For a user whose basic work tool is spreadsheet computation, the study by Eaton et al. (2019) provides no help for computing confidence intervals.

**1.2 reply by authors**

This is very useful feedback for us. Our intention is indeed to provide a user-friendly tool that implements binomial statistical theory to calculate confidence bands about grain size distributions to prevent type 1 statistical errors. The revised manuscript now provides an overview of the confidence interval calculation procedure, and then lays out the precise statistical basis for the calculations for different kinds of data (i.e. raw observations and binned data). We have also written a new pair of functions to perform sample to sample comparisons to determine whether sample grain sizes for a percentile of interest are statistically different.

We have re-written the introduction and statistical basis sections of the paper, and we have added an overview section to better explain

1. how the binomial distribution can be applied to both raw data comprising $n$ measurements of b-axis diameters and also to the typical binned data collected in the field; and

2. how the binomial theory can be used to generate confidence intervals about an estimate of a given grain size percentile.

The process is summarized in the new overview section, which describes how to estimate the percentile confidence interval (a term we introduce and use throughout the revised paper), and how to map that onto the sample cumulative frequency distribution to estimate the associated grain size confidence interval. The distinction between these two things is at the root of much of the confusion generated by our original manuscript.

In addition, we have written an appendix to the paper that describes how to use the simpler normal approximation to the binomial distribution to calculate the confidence interval, as well as a spreadsheet implementation of that approach.

We have also created an appendix containing reference tables of the percentile confidence interval bounds for a range of percentiles of interest (i.e. $D_{10}$, $D_{15}$, $D_{20}$ ... $D_{90}$), sample size ($n$), and acceptable confidence limit ($\alpha$).

**1.3 Comment 2**

Computation of confidence bands around grain-size distributions without assuming an underlying distribution type is not a new idea. Fripp and Diplas (1993) presented a binomial approach to compute the relation between sample size and error around individual percentiles. The study by Church and Rice (1996) applied a bootstrap approach to a large pebble count of 3500 particles and computed error bands around various percentiles of the grain size distribution. The grain-size distributions did not fit a particular distribution type, but the bootstrap confidence limits were reasonably close to those computed assuming an underlying skewed log-normal distribution. Petrie and Diplas (2002) cautioned that ...the binomial distribution considers only two possibilities for each particle sampled: (1) the particle is within a specific size class (e.g.,smaller than a certain size) or (2) the particle is not within the specified size class. The binomial distribution is then inadequate to use for representing entire size distributions. To overcome this limitation and to compute confidence bands around the cumulative frequency distribution from a pebble count with data binned into size classes while considering distribution characteristics of the distribution, Petrie and Diplas (2000) developed a multinomial approach.

**1.4 reply by authors**

This is also very useful information for us, and we have read the papers with interest. The work by Diplas and colleagues is particularly relevant and strengthens our paper. The analysis by Fripp and Diplas (1993) is now used as a jumping off point for our analysis: we have re-written our manuscript to use that paper as the basis from which we start, we describe that approach in the appendix, and we have implemented a version of it in a spreadsheet that accompanies this paper.

The paper by Rice and Church (1996) was the inspiration for the re-sampling analysis that we presented in our original paper. However, we have clearly not done justice to the analysis presented therein, so we have expanded that section.

The work by Petrie and Diplas with multinomial theory is primarily focused on determining the sample size required for a given level of accuracy for estimating the shape and relative position of the cumulative grain size distribution, using binned data. Our approach and intent is different: we develop our statistical theory using individual measurements of b-axis diameters, and we develop confidence bounds to be plotted when comparing distributions to avoid type 1 and 2 statistical errors. In this context, the binomial approach is most appropriate. Our implementation of binomial theory is based on the interpretation that a measured stone is either (a) greater than a percentile of interest for the population, or (b) less than or equal to the percentile of interest, with no reference to or limitation imposed by having binned data. In this context, the estimation of $j$ percentiles involves the execution of $j$ independent binomial experiments with assumed probabilities corresponding to the percentile of interest. To test the difference between our approach and the traditional binned data, we use the scheme described in our paper to directly compare the distributions based on all measurements, and the binned data.

While current practice in the field is still to collect binned data, the automated techniques for grain size analysis that are standard practice in most experimental laboratories, and which are being increasingly deployed in the field promise to deliver much more data than can be collected manually, and will obviate the need for binned data. Our methodology is best leveraged in that context, using the automated data analysis approach possible using languages like R and Python. Therefore, our differentiation between binned data and the underlying b-axis diameter measurements is not simply a technical one, it is based on our perceptions of the future data types that will be commonly used.

**1.5 Comment 3**

While the study presented by Eaton et al. (2019) is successful in raising awareness that the n=100 sample size is too low to attain reasonable accuracy for pebble counts in most gravel beds and that sample sizes of 400 or 500 particles are required to enable statistical evaluations about sameness or difference, the study does not succeed in presenting its computational approach in an easy to understand way. Providing

computer code in R-language is not helpful for most users, hence the authors computations cannot be repeated or applied by users who are not expert statisticians but are seeking to determine confidence limits around their sampled grain-size distributions.The authors display the confidence bands that they drew with their binomial approach around grain-size distributions sampled in other studies (Kondolf, 1992; Bunte et al.2009, Bunte and Abt, 2011) and go on to discuss whether the now-drawn confidence bands warrant the interpretations made in the original studies. In the final sections of the study, the authors show general relations between sampling error, as computed with their binomial approach, and sample size as well as distribution sorting.

**1.6 reply by authors**

We are very grateful for the feedback about the relative difficulty in understanding our approach, and about the need for addition means of implementing our tools for estimating the confidence bands. We have responded to the first point by re-writing the section of the paper presenting the method, and to the second by developing reference materials in two appendices, as well as a spreadsheet implementing the normal approximation to our solution, as described by Fripp and Diplas (1993).

**2 Recommendations for improving the paper**

The reviewer made several helpful suggestions for improving the paper, listed below:

> Reference prior work and build on it Eaton et al. (2019) should discuss prior studies that likewise compute errors around percentiles without assuming an underlying distribution type and explain the improvements and advantages offered in the study presented. What reason is therefor a user to select the authors approach if the authors do not explain WHY their approach constitutes an improvement?

We have improved the links between our paper and the previous work. We also re-iterate in the revised paper that our main purpose is to produce a user-friendly introduction to the basic method for estimating confidence bounds using binomial theory. We point out that our approach is statistically conventional, has precedents in the literature, and is consistent with empirical analyses. We also more strongly articulate our key message – that all grain size curves ought to be plotted with confidence intervals, particularly when two distributions are being compared.

> Provide explanations and instructions In order for readers to apply the binomial approach to their own data, the authors need to provide a step-by step explanation on how to use their approach rather than referring to a book on statistics, pointing to a website, and offering computer code in R-language. Offering a reader access to computer code is a courtesy, but not a substitute for a step-by step explanation, especially not for a very hands-on and applied topic of monitoring bed-material changes.

With this particular comment in mind, we have re-written the manuscript and generated various reference materials.

> Comparison of results to those from prior work: How do percentile errors computed from the authors binomial approach compare to percentile errors computed from other approaches? Apart from a similarity of sampling errors around the D50 and D84that the authors computed from their binomial as well as a bootstrap approach for asymmetrical grain-size distribution (the authors flume experiment), the authors do not show how their binomial approach to computing confidence bands relates to confidence bands computed from other approaches. The authors should apply their binomial approach together with the approaches suggested by Fripp and Diplas (1993),Petrie and Diplas (2000), and Rice and Church (1996) as well as simply to sample-size equations for an error around the mean to a few pebble-count distributions that differ in their sorting and skewness (esp. the extent of a fine tail) and then assess difference sand similarities between results.

In our revised paper, we make the links to the cited literature clear, and we replicate the approach described by Rice and Church, and then compare it to the binomial methodology we describe.

> Explain whether or how confidence intervals computed from the binomial approach are affected by sorting and skewness of a sampled grain-size distribution While the authors show that confidence bands increase in width with a distribution's sorting co-efficient, the authors do not explain how exactly sorting (and skewness) of a sampled grain-size distribution (e.g., a tail of fines) flow into the computation of confidence

intervals based on the binomial approach. The binomial approach introduced by Fripp and Diplas (1993) does not seem to involve sorting or skewness of the sampled distribution, suggesting that confidence intervals from a binomial approach are similar for all percentiles within a sampled grain-size distribution with a known sample size and number of size classes.

The revised text and several new figures address this point.

Have a user in mind and offer a procedure that is reasonably easy to be applied by the user The authors provide a study that is of interest to users who are involved in relations of sample size to error. However, the study is geared towards a statistically expert audience rather than the needs of non-expert potential users. If the authors' work is to be applied for monitoring purposes by staff from environmental agencies or consulting and by those whose main interest is not statistical but who need to apply such relations, then the authors need to provide detailed explanation and instruction.A spreadsheet implementation of their computations of a percentile error would be considerably more helpful than code in R-language.

We have developed additional resources that address this point, and we are particularly thankful for this feedback, since our main purpose is to make it easy for people to use our approach.

Editing suggestions Figures provided by the authors are generally fine, but considering that the study discusses plotted details of whether or not confidence bands overlap,a larger figure size would be helpful. It would also be helpful to place the figures below their first mention in the text, not simply at the top of the page with a mention some-where below on the page. With respect to writing style and typos (etc.), the manuscript is well written and clean

We have re-worked many of our figures, but will leave it to the editorial staff to properly place the figures in the final version of the manuscript.

**3 Reviewer 1: Specific comments**

The reviewer also provides a list of specific comments that improved the paper. Those comments are quoted below, along with our responses to them.

p.2, l. 15: ". . .but the largest source of uncertainty in many cases is likely to be sampling variability, which is a function of sample size." How do the authors know that sampling variability (do they mean statistical uncertainty due to a poorly sorted channel bed?) rather than methodological differences (e.g., measurements of particle sizes, spatial heterogeneity, differences in the sampled channel width or leaving poorly accessible stream locations unsampled) is the most likely factor causing uncertainty? The comparative study by Bunte et al. (2009) showed that differences in sampling outcomes due to methodological variability can be huge.

In order to avoid confusion, we have rewritten the sentence to read "but the largest source of uncertainty in many cases is likely to be associated with sample size, particularly for standard pebble counts of about 100 stones."

p. 2, line 21: ". . . We then use this approach to demonstrate that the higher percentiles, such as D84, are subject to substantial uncertainty for typically used sample sizes, and that. . ." 1) Given this statement, it is odd then that the confidence bands drawn by the authors around the size-distributions from two streams sampled by Bunte et al. (2009) and another stream sampled by Bunte and Abt (2001) are all narrower for the D84 than for the D75 and the D95. 2) That statement is not backed by results from other studies: Rice and Church (1996) have shown for a very large pebble count that uncertainty was lowest for the D75 and D84 sizes, followed by the D50 and D95 sizes, and highest for the smallest percentiles. Green (1993) corroborated this finding; on average, the D73 could be determined with the least uncertainty. Similarly, Bunte and Abt (2001a) found in their field study that uncertainty was lowest for the D50 and the D75, slightly higher for the D84, D95 and D25, and percentiles lower than D25 were subject to the highest uncertainties.

This section of the paper has been re-written, and this sentence is no longer included. The underlying issue that that the standard graphs use lognormal axes. As a result, the uncertainty expressed in mm for the D84 is in fact larger than it is for the D50, even when the uncertainty expressed in phi units is smaller. The point is not an essential one in any case, and is no longer relevant, given the revisions we have made.

> p. 3, l. 3: ". . .since we preserve each measurement rather than grouping them into size classes, the data can be treated as a binomial experiment, . . ." Does that mean that the binomial computations is not applicable to field data binned in 0.5 phi units which results from measuring particle size using a 0.5-phi template?

We have hopefully addressed this question more clearly in the new section presenting an overview of the method we use and in the revised section where we discuss how to apply binomial theory to binned data. (in any case this section containing this sentence has been rewritten to improve clarity).

> In Eq. 1, Pr and p are not defined

This equation is now introduced (and defined) in the overview section to improve clarity. It is used first in an example of the standard coin toss binomial experiment, and then in the directly analogous problem of estimating the bed surface $D_{50}$.

> p. 4, line 10-19: The description of the methodology is too vague. To allow a reader to replicate the computations, authors need to provide step-by-step guidance. Reference to websites and other studies is not sufficient for a paper that would like to introduce a new approach to computing confidence bands.

The new overview section and the re-written statistical basis section hopefully address this point. We have also adopted the term *percentile confidence interval* and *grain size confidence interval* throughout the text to more clearly explain how binomial theory can be used address the uncertainty associated with sampling (i.e. the percentile confidence interval), and how the shape of the cumulative frequency curve determines the uncertainty for a given grain size percentile estimate (i.e. the grain size confidence interval). In the overview section, we use a new figure to explain the relation between percentile and grain size confidence intervals.

> p. 4, line 21: That statement comes out of the blue . . .what areas? What tails?? Fig. 2 does not provide much help either.

The sentence now reads "One disadvantage of the exact solution described above is that the areas under the tails of the binomial distribution differ". The Figure shows the binomial distribution, so the link between the figure and the text is now more explicit. We have also modified the figure caption and legend labels to explicitly identify the distribution tails.

> p. 5, line 1-5: Again, step-by-step instructions are needed to allow a reader to replicate the authors' approach.

We have tried to address this confusion by creating the overview section that precedes the admittedly rather dense description of the statistical basis for our approach. The precise mathematical approach is laid out in the code behind our functions in the `GSDtools` package (note: we have changed the name of the R package to reflect its more general nature since the addition of two hypothesis testing tools); the underlying calculations which are described in the text can be viewed mathematically by installing the package and then typing `WolmanCI` at the command line prompt. We have also included the source code for the functions in the online archive of code and data associated with this paper. The purpose of publishing an R package is to make our exact code and methodology available for both scrutiny and practical use. We have implemented the simpler normal approximation used by Fripp and Diplas (1993) in a spreadsheet version, and we have described the basis for this approximation in a new appendix to the paper. Hopefully, these additions will help potential users replicate our approach.

Also, we have included step-by-step instructions for the two new functions we have created to test hypotheses about differences between two samples.

> p. 6, line 5-6 ". . .Based on the overlap in 5 confidence intervals for the eight samples, the distributions do not appear to be statistically different (see Fig. 3). . .. 1) Confidence bands plotted by the authors for their stream table sediment overlap for samples 2 and 3, but not for samples 1 and 4 (Fig. 3, panel A). 2) With respect to their multinomial approach, Petrie and Diplas (2000) stated that error bands are identical for all particle- size distributions as long as the value for alpha (e.g., 0.05) and the number of sampled size classes remain the same. For the authors' 8 samples from the stream table sediment surface, I assume that the same number of size classes were collected in each of the 8 samples and that the same alpha value was applied to all computed confidence bands. If the statement by Petrie and Diplas (2000) was true for the error bands conducted by the authors, then why do the error bands plotted in Fig. 3

differ between samples? 3) The authors use as basis for their analyses a sand-rich sand-gravel mixture with a D50 near 1.5 mm. The lengths of b-axes appear to have been determined to a precision of two decimals (e.g., 0.53 mm). It is difficult to imagine how a pebble count was performed and particle sizes were measured on sediment this small.

This section has been completely re-written, and the text and figure referred to has been removed. In summary though, the data collected were not binned into size classes, individual grain diameters were recorded; the error bands referred to by Petrie and Diplas are percentile confidence intervals, not grain size confidence intervals (an issue we explain in our new overview section); and the measurements were made from a digital photographs of the bed taken 15 cm above the bed with a pixel resolution of about 50 microns. Obviously this introduces the possibility of grains being partially hidden in the photo, but this effect is far less pronounced in laboratory sediments because, due to scaling issues, sediment finer than the field equivalent of 10 mm grains are not included in the bulk mixture (i.e. there are relatively few 'fine' grains that can fill in pores and obscure the larger grains the way they can in the field). In addition, the purpose of these data is simply to represent a known population of grains from which to draw samples, not to actually represent the bed surface GSD of the experiment accurately.

p. 7, Fig. 4: 1) While the box of box and whisker plots typically shows the quartiles, there is less standardization of what the whiskers represent. Please indicate what the whiskers in this plot represent. It can't be the overall spread because "outliers" are plotted as dots. Please define. 2) What parameter is plotted on the y-axis? Please clarify. 3) It would have been useful to show the 95

We have abandoned this figure, and instead used a different approach to test the binomial predictions against bootstrap error estimates for a much wider range of percentiles. The new figure plot the predicted and bootstrap errors on a typical grain size distribution curve, and we evaluate their goodness of fit using s 1:1 model (i.e. a model of perfect agreement) and the Nash Sutcliffe goodness of fit statistic (which is basically the same as an $R^2$ value, where 1 equals a perfect model). The completely re-written section on confidence interval testing now engages with previous approaches more explicitly and is more extensive. Note that we replicated the entire confidence interval testing using a different population of grain sizes defined by 1,000,000 observations drawn from a log normal distribution with virtually the same results.

p. 6, line 9-19. The authors state that they found a close match between the confidence bands computed from the binomial approach and a bootstrap approach (Fig. 4) for an unskewed grain-size distribution (i.e., their stream table sediment). The comparison plot by Petrie and Diplas (2000) for a pebble count from the Mamquam River shows that the confidence bands computed with the approach by Fripp and Diplas (1993) are between n 0.02 and 0.06 phi-units higher than those from the bootstrap approach computed by Rice and Church (1996). Is the binomial approach by Fripp and Diplas (1993) similar or different to the authors' binomial approach? Does a binomial approach yield wider confidence bands than a bootstrap approach?

We address all of these points in revisions to the introduction (where we talk about the Fripp and Diplas approach), and in the confidence interval testing section. We write in the revised paper "The advantage of a bootstrap approach is that is replicates the act of sampling, and therefore does not introduce any additional assumptions or approximations. The accuracy of the bootstrap approach is limited only by the number of samples collected, and the degree to which the individual estimates of a given percentile reproduce the distribution that would be produced by an infinite number of samples." The differences observed by Petrie and Diplas are presumably due to their use of the normal approximation of the binomial distribution.

p. 7, lines 11-19: In the authors' reassessment of particle-size distributions from Kondolf (1997) and from Bunte et al. (2009), the authors need to clearly state to what percentage confidence the plotted confidence bands refer? I assume they are 95% confidence bands. Please clarify.

Figure captions all now clearly indicate that the polygons represent 95% confidence intervals.

p. 8, Fig. 6: The study by Eaton et al. (2019) has drawn confidence limits around grain- size distributions from three Rocky Mountain gravel-bed streams sampled by Bunte et al. (2009) and Bunte and Abt (2001). 1) Based on visual examination of the error bands plotted in Fig. 6, I'd say that for Willow Creek, the error bands for riffles and pools are different except for the narrow range between 20 and 50 mm within which they cross. 2) The plotted confidence intervals for Willow Creek and the St. Vrain are jagged around the sampled distribution and seem to widen notably for the flatter sections of the cumulative size distribution but neck down for the steeper sections. The authors offer no explanation for this phenomenon.

The observed changes in the width of the grain size confidence interval do indeed correlate with the shape of the cumulative frequency curve. This effect is due to the mapping of the percentile confidence interval onto the grain size confidence interval. We have added a new figure and an overview section to better explain this point. Comparing samples to determine whether a given percentile of interest is different or whether the samples can be considered different as a whole can only be approximately done using a visual interpretation of the confidence intervals. We have developed two new functions to rigorously compare samples; these functions (and the step-by-step instructions for how to conduct the analysis) are presented in the statistical basis section; they are also used in the reanalysis section; and they are included in the online demonstration of how to use the GSDtools package.

> p. 10, line 9-14: The authors write: "Our method for estimating uncertainty requires only the cumulative distribution and the number of measurements used to construct the distribution. Therefore, confidence intervals can be constructed and plotted for virtually all existing surface grain size distributions (provided that the number of stones that were measured is known, which is almost always the case),. . ." If computation of the width of the confidence interval for any percentile of interest re- quires only knowledge of the sampled distribution and sample size n, and if the computation is conducted for each percentile individually, then how does the spread or sorting of the sampled distribution influence the computed confidence interval? Please CLARIFY!

This is explained in the overview section, and relates to the difference between the percentile confidence interval and the grain size confidence interval.

> p. 11, Fig. 9 and p. 12, Fig. 10: 1) The units in which the error is computed needs to be clearly stated. Somewhere down in the text the reader gets a hint that the error pertains to a percentage error in mm units. 2) The findings that percentile errors decrease with sample size and with the distribution sorting is in and of itself nothing new. What is new here is that the error is computed from the authors' binomial approach (assuming an underlying log-normal distribution for Fig. 10). To allow a reader to see whether there is a difference between errors computed from the authors' binomial approach and other approaches (e.g., Fripp and Diplas (1993) or simply errors around a mean), the computed relations between errors and n should be compared to errors computed with other approaches. 3) For comparison with other studies that compute percentile errors in terms of absolute +- error in phi-units it would be helpful if the error-n relations in Fig. 9 had a second y-axis with error in terms of the absolute +- error in phi-units. 4) It would be useful if the relation of error to n was also provided for the error around the D16.

The intention of this section is to provide the user with some guidance related to sample size required to reach a specified level of precision. As should be now clear in the revised paper, the grain size confidence interval cannot actually be estimated until the sample is collected. As a result, we have compared our results to those from others in the confidence interval testing section. This section has been edited to better emphasize that the analysis is only meant to guide sample size estimation, but does not obviate the necessity of calculating the grain size confidence intervals once the sample has been collected. With respect to the units, Eq 3 is now written so as to make it clear that we are calculating a normalized difference, which is by definition dimensionless.

> p. 12, line 8: The authors state that for a given n and sorting, errors are largest for steep gravel fans and bar top surfaces and smaller for typical gravel beds with a sorting near 1. That is a useful comment. It would be even more useful to elaborate a little bit here on what kind of sorting values to expect for different morphological or sedimentary channel units and hence what a user needs to expect in terms of the error - sample size relation.

We agree with this comment, which is what motivated us to model the effect of grain size distribution spread on uncertainty using log normal grain size distribution (the following section). Unfortunately, our data do not support even finer resolution of the issue on a sedimentary unit by unit basis.

> p. 14, line 12-13: I am afraid that the authors' time estimates refer to dry deposits of mainly mid-sized gravels. The time requirements for a 500-particle pebble count in- creases to about 5 hours when sampling in poorly wadeable conditions, in the presence of abundant algae and large woody debris, under overhanging bushes, and with particles being next to irretrievable from the bed because they are tightly wedged within neighboring particles or small particles placed in tiny pockets between large clasts. The necessity for a large sample size remains, but users and their funding agencies need to commit to realistic time requirement.

We have incorporated the reviewer's time estimate for more arduous samples in a sentence that reads " In less ideal conditions or when working alone, it may take upwards of 5 hours to collect a 500 stone sample, but as we have demonstrated, the uncertainty of the data increases quickly as sample size declines (see Figs. 10 and 11), which may make the extra effort worthwhile in many situations."

Typos etc. p. 2, l.5: The value should be 22.6 (=2ˆ0.5*16), not 22.7. p. 3 L. 5. . .compute the quantiles of the (Fig. 1). Something is off in that sentence. p. 4, Footnote: The access date is in the future.

We have fixed all of this smaller issues.

**4 Reviewer 2: General Comments**

The comments provided by Reviewer 2 are presented below, along with our responses. Many of the points have been addressed in our reply to Reviewer 1 above, but these comments were equally helpful in re-shaping the paper, particularly in those instances when Reviewer 2 has identified the same points raised by Reviewer 1.

**4.1 Comment 1**

The submitted paper focuses on estimating uncertainties in measured grain size distributions using statistical analysis of grain size data from experiments, field measurements and synthetic data. I think that the authors make an important main point, which is that uncertainties in grain size distributions should be reported especially when used to assess grain size changes over time or in space. Although I am supportive of the overall goals, topics, and messages of this manuscript, I think that there are many details missing from the methods. This makes it difficult to evaluate how this calculation is actually applied, the assumptions involved, and finally how it compares to previously published studies on uncertainties in grain sizes. I suggest adding these details such that your paper can be understood by a broader audience.

**4.2 reply by authors**

To address these concerns, we have re-written much of the paper and generated additional figures that we hope better describe how our approach actually works. The revised paper also includes an expanded results section that clarifies the links to previous work, as well as reference appendices providing supporting information. We also now provide a spreadsheet that implements the normal approximation to our technique (as described by Fripp and Diplas, 1993) to estimating percentile confidence limits. Finally, we added two functions for explicitly comparing two samples to determine whether differences in the grain size estimate for a given percentile are actually significant. We appreciate all of the suggestions that are made in this review, and we are confident that the revised version will reach a broader audience.

**4.3 Comment 2**

I would really like to see a more detailed review of what previous studies have done to quantify uncertainties in the D50 and other percentiles of the grain size distributions. Do approaches without an assumed grain size distribution exist? If so, what is wrong with these approaches that motivates this current study? I'm a bit confused because in the introduction you state that there is no easy way to estimate the required sample size. In the abstract you also write that you propose a simple approach to estimate sample size, but this also relies on assuming a log-normal distribution as in previous studies highlighted on p 2 lines 8-9. What is the difference between your approach that assumes a log normal distribution to estimate sample size and other log normal approaches? It is not entirely clear to me in reading the introduction what is new in this study compared to previous approaches. A more in depth review of previous approaches and a statement of how this new approach is different would really help.

**4.4 reply by authors**

We have extended our discussion of previous approaches by re-writing the paper to leverage the previous work by Diplas and colleagues as the starting point, and we describe in more detail how we replicated the bootstrap approach of Rice and Church to estimate the uncertainty of samples with various sizes drawn from our population of 3411 b-axis measurements. Basically, we believe that our approach is entirely consistent with that proposed by Fripp and

Diplas (1993), and replicates the empirical results presented by Rice and Church (1996). The main issue that we try to address in this paper is not that previous methods are flawed, but rather that we as a community have failed to use those approaches to quantify sampling uncertainty (despite the precedents in the literature). As a result, there are published results that are clearly not statistically defensible, and it is our impression that many people continue to collect relatively small samples with limited appreciation of what that means in terms of uncertainty.

In our revisions, we will also emphasize that we think are the main contributions of this paper, which are:

- to describe clearly how surface sampling can be described as a binomial experiment, analogous to a traditional coin toss experiment;

- to present a simple set of tools based on binomial theory with which anybody can easily calculate the grain size confidence interval about any sample percentile that will contain the population percentile size;

- to demonstrate the importance of considering uncertainty when comparing samples of the bed surface, or when making calculations based on those samples; and finally

- to make some assumptions about distribution shape so that we can provide some general guidance on the sample size required to reach a desired level of sampling precision.

This last point involves making assumptions about the underlying distribution (i.e. we assume a log normal grain size curve), but that is simply to generate synthetic data with which to model the effect of sample size and the spread of the distribution on the precision of a percentile estimate. We will make it clear that any distribution form could have been used, but that we chose a log-normal distribution because (1) it is the simplest to describe (i.e. it can be described by a mean and standard deviation), (2) it has been used previously by others, and (3) many gravel beds are approximately log-normal. We more clearly emphasize our central message in our revisions, and de-emphasize the point about sample size.

**4.5   Comment 4**

The reviewer made several comments about our calculations that we would like to address:

> In section 2.1, how is equation (1) used? Please provide a step wise explanation nohow someone would perform these calculations and what information is needed. Right now it is somewhat difficult to understand how equation (2) is actually solved. Although I appreciate the inclusion of the R code that is part of this paper, a simple explanation of your detailed methodology is really needed in the main text to properly evaluate your methods. What are successes, please define. I am also somewhat confused about the definition of p, earlier you state it is the percentile of a distribution but on P 4 L6 is it called a probability.

We have completely re-written the statistical basis, including an overview section that walks the user through the idea of a binomial experiment, the probabilities of a particular outcome (and the relation of those probabilities to the grain size percentiles for the population being sampled), and the relation between percentile confidence intervals and grain size confidence intervals.

> In section 2.2, please also provide more details on this approach, one brief sentence on interpolation really does not make this calculation clear.

We have re-written the entire section to improve clarity.

> Section 3 and Figure 4 How many times did you create a sample with 100 grains to make these distributions in Figure 4? It seems like the results could really vary with the number of 100 grain samples? Also, some explanation of the boxplots is needed to evaluate the results. What are the horizontal lines at the top and bottom ends of the distributions? This information is needed to validate that the two predictions actually provide similar results. Can you provide the actual numeric values of the 99%confidence interval bounds for the two methods in the figures to enable quantitative comparisons?

We have re-written the entire section with these comments in mind. We repeat the kind of bootstrap error estimates presented by Rice and Church (1996), and make a more extensive comparison of the binomial predictions and the bootstrap estimates. We ended up taking 5000 samples from the population to ensure that the distributions of estimates stabilized. In addition, the entire analysis was repeated using samples from a synthetic log normal population of 1,000,000 observations; the re-analysis yielded nearly identical results.

**5 Reviewer 1: Specific comments**

The reviewer also provides a list of specific comments that will improve the paper, listed below, followed by our response to them.

> P 1 L 21-22 For facies mapping, my understanding of the Buffington approach is not that it is meant to be purely qualitative as implied here. They have visual classification of patches that are then verified by numerous pebble counts on the patches. So their approach likely provides a more accurate representation of the grain size distribution because they use many pebble counts in a single reach.

This is a good point, and we now refer to semi-qualitative methodologies to avoid the issue.

> P3 L5 Missing word(s) here.

This text has been deleted during the revisions.

> P4 L 12-16 Please state if this text is for a specific sample (e.g. the data shown in Figure 1), right now it seems to be written as if it applies to all grain size measurements but I don't think that is actually the case?

It is in fact true for the percentile confidence interval for all samples, but not the grain size confidence interval (which depends on the shape of the cumulative frequency distribution for the sample in question). We have made extensive edits to existing sections and we have added an overview section that addresses this point explicitly.

> P4 L 15-16 Please explain what you mean by 19 times out of 20. I'm not clear why these exact numbers are chosen instead of a percent of trials. It is also not clear how this percent of trails was calculated or how the range of 159-180 was determined.

This section of the paper has been re-written and is augmented by the new overview section that now better explains the how the bounds to the percentile confidence intervals are determined.

> P4 L 21-23 Stating that the area under the tails differs is pretty vague. Do you mean tails of the distribution? How are the tails of the distribution defined? Please state why these different areas are problematic. Similarly, upper and lower limits of what exactly? What do you mean by a one-sided interval and how does this relate to your calculations? I can guess what you mean but the lack of language specificity here makes your text somewhat difficult to follow.

This section has been re-written to improve clarity, and is also augmented by Appendix A, which describes how the confidence interval bounds are determined using the more familiar normal approximation to the binomial distribution. We have also added text to the caption of the figure that explicitly references the distribution tails. We also define one-tailed distributions (though admittedly it remains a technical, statistical definition).

> Figure 3 More details are needed as to how the grain size data were collected, through a random sample or grid count? Were the samples in different locations on stream table and using the same or different operators? It is a little difficult to see the confidence bounds in this figure to assess overlap of various distributions, not sure though how you can easily address this problem.

This figure has been deleted during the re-write of the paper. The main point is that we have a population of 3411 measurements that we can use to replicate the bootstrap error calculations performed by Rice and Church (1996). Since the time and space distribution of the sub-samples used to generate this population is never referred to in the rest of the paper, we chose to delete the figure and simplify the text. Where this population is first introduced, we provide a bit more information about the sampling, as requested. The sentence in the paper reads "the population shown is defined by 3411 measurements of bed surface b-axis diameters at randomly selected locations in the wetted channel of a laboratory experiment performed by the authors."

> P 7 L 8 typo here

Fixed

Figure 5. I appreciate this reanalysis but I don't think that you can say that the distributions are statistically similar or different without a similar confidence bound on the bulk sample data. Previous studies have demonstrated that bulk samples also have considerable uncertainty depending on the size of the actual bulk sample and the portion of the sample that is occupied by the largest grain sizes. So the bulk sample is also not free from uncertainties and this needs to be acknowledged.

This is a fair point. Given that we have added new sections and figures to the paper, and that we have extended our comparison of our method to previous methods, we chose to remove this figure and the associated analysis.

P 8 L 3-5 The statement that fine sediment would be deposited preferentially in the pool rather than in the run/riffle during the waning limb of the preceding hydrograph needs some references to support it.

We have added references to some of the seminal work on this topic.

P 12 L 6-7 Please explain why you are assuming the standard deviation of the distribution is related to logD84-logD50.

To make the paper clearer and to improve the comparability of the field data and the results of the log normal simulations, we now use a sorting index ($\phi_{84} - \phi_{16}$) to quantify the spread of the distribution. This is, we think, a clearer way of conveying what we did without introducing unnecessary complications.

P 12 L 10-12 I do not entirely why you are simulating log-normal samples with this given range of D50 values and SDlog values? How were these distributions simulated by defining D50 and SDlog beforehand? Figure 10 does not seem to be referenced or explained anywhere in the text.

We have added edits at various points in this section to make a few points related to this comment. We point out that the purpose of this section of the paper is simply to provide some guidance to choosing an appropriate sample size, and that this is a secondary objective of the paper (the primary objective being the articulation of the importance and relative ease of generating confidence intervals about bed surface grain size distributions). We also now clearly state that we approach this problem first using a set of field data to estimate the grain size uncertainty associated with different sample sizes, and second by using log normal distributions to quantify the effect of data spread, indexed by standard deviation. We generated the log normal distributions using the `rnorm` function in R (e.g. `GSD = 2^rnorm(n = 352, mean = 5.6, sd = 1.3)`).

P 13 L. 14-22 More details are needed as to how you estimated that this grain size is entrained at a certain shear stress and discharge. Did you use Shields equation? What critical Shields stress did you assume? How did you then translate this shear stress into a discharge beyond using a stage-discharge relation; did you have a measured channel bed slope and are you assuming stage is equivalent to the average flow depth in a reach? What is the basis of the assumption that D50 becomes fully mobile at twice the shear stress needed to initiate D50 movement? Some rational and supporting references are needed to support this argument. I am also a little confused about this uncertainty in grain size because all of these sizes (46, 55, 64 mm) are essentially in the same half-phi bin. I may be mistaken but if you have binned your data into half phi intervals for this analysis, wouldn't you expect a similar, although likely smaller, level of uncertainty in the D50 anyway? This uncertainty would occur because you are deter- mining the measured stream bed D50 value (55 mm) by interpolation between the two percentiles straddling the 50th percentile value, and these two bounding percentiles correspond to grain size bins 45 and 64 mm. But you do not actually have any grain size resolution finer than half phi bin size. So when you calculate a median grain size of 55 mm, you are interpolating this grain size to a finer resolution than you actually have data. Doesn't this already seem to imply that your uncertainty in D50 might be somewhere within a half phi bin size when you only have binned data, depending of course on how the actual grain sizes are distributed within that half phi bin?

We now explain how we determined the entrainment threshold (visual observation of painted tracers, confirmed to occur at a dimensionless shear stress of about 0.045). The other details of the methodology to estimate shear stress are described in the referenced papers. We have added a reference supporting full mobility at twice the entrainment threshold. The issue of interpolation using binned data, and the accuracy of that kind of data relative to individual measurements of b axis diameters is now addressed in the overview section and in the re-written statistical basis section. In particular, our new Fig. 3 demonstrates that the differences between binned data and interpolations from cumulative data are small compared to the sampling confidence interval, which means that, in practice, binned data can be treated as if they were not binned.

P 15 L 12-13 Although I certainly agree that having more than 100 sampled particles would be better for uncertainties in most studies, these time estimates assume a team of people performing pebble counts. Having conducted a very large number of pebble counts on my own, these can take much longer than 20 minutes. The time also really depends as to whether you are binning grain sizes or measuring individual b axes. Finally, setting up and finding grains on a grid also adds to the pebble count time, so I would argue that this 20 minute estimate is a minimum.

We have added a note to this section of the paper that does acknowledge the difficulties of collecting large samples in arduous conditions.

[revised manuscript text omitted]

---

## Author Response (AR3)

**Estimating confidence intervals for gravel bed surface grain size distributions: Reply to reviewer comments**

Brett C. Eaton[1], R. Dan Moore[1], and Lucy G. MacKenzie[1]

[1]Geography, The University of British Columbia, 1984 West Mall, Vancouver, BC, Canada

July 19, 2019

**1 Reviewer 2: General Comments**

The second round of comments from Reviewer 2 are presented below, and are discussed at some length. We attempt to address all the key issues, and highlight how we have responded to them.

> P2 L24 I would have to agree with Kristin Buntes original review about this line in the text. It is very difficult to say what is the largest source of uncertainty in measured grain size distributions and all of the other things that she lists could be just as important as sample size. For example, not choosing an appropriate representative sample location in a reach (with wide spatial variations in grain sizes) could lead to larger errors than not sampling enough grains in the right location. I appreciate the authors changes to the text but I think that this sentence is still misleading and ignores many other error sources. I would suggest simply changing the sentence to ...is ONE of the largest sources of uncertaintIES... to address this.

We have made the recommended change.

> Section 2.1.3 on approximate solution. I think it would be good to note here, beyond the title of this section, that this relies on assuming equal area tails. Doesnt this mean that the more the grain size distribution deviates from the equal area tail assumption, the larger the errors in approximate solution? If so, a note of caution is really needed for potential users of this approximation.

Appologies for the confusing wording: the distribution in question is the binomial distribution which describes percentile uncertainty. No assumptions about the distribution of the grain size is made. Therefore the equal area tail approach can be applied to all grain size distributions regardless of shape. We have changed the subsection heading to make this less confusing. The purpose of the method described in Section 2.1.3 is to produce symmetrical confidence intervals by interpolating from the binomial distribution. So the heading now says that.

> Section 3.1 I think that in step 3 of the bootstrap approach, you need to clarify that you are calculating the desired percentile value from the bootstrap samples, rather than just the samples as stated here. The original data used in the bootstrapping approach were also called samples so this step may be confusing as to which exact samples are being used here (original or bootstrapped data). Also shouldnt the subscripts in the equations for this step be xk rather than x and yl rather than y? I would recommend the same changes are made in Section 3.2, which has similar steps, equations, and use of the word sample in step 3.

We have re-written these to sections, as suggested. The re-written text should make it clearer why the subscripts are written as they are.

> Figure 5. I am somewhat confused as to why the 95% confidence interval bounds seem to be different between Figure 5a and 5b. If the confidence bounds are calculated from the true population of 3411 measurements, then shouldnt they be identical between the two figures because they are independent of the sample size of the randomly chosen grains? If the confidence bounds are not based on the entire population of rocks, can you please explain this more in the text because as currently written you state

along the with 95% grain size confidence interval bracketing the true grain size population. Also, why do the confidence intervals not extend though the entirety of the distribution (the tails of the distributions are missing confidence bounds)?

We have added text to the manscript explaining both of these results. The computed confidence intervals use the population grain size distribution, but sets the sample size to 100 and 400, respectively. That way they are directly comparable to the samples of 100 and 400 stones taken from the population, and are different from the confidence intervals that would be produced using $n = 3411$ (that confidence interval is every narrow). We also observe that researchers almost never make use of percentiles outside the range D5 to D95, and therefore we end our confidence intervals there (this also happens to be the limit of applicability for the normal approximation to the binomial distribution used by Fripp and Diplas, for a sample size of 100, and is thus a good practical limit to use). We also adopt the same convention for all subsequent plots showing the confidence intervals on a graph (a change from the last version), and state this convention in the manuscript at this point.

Figure 6. Please see my comment in Figure 5, why do the confidence bounds not extend to all grain size percentiles and seem to end at about 5 and 95 percentiles? I think there is also an error in labeling this figure (three of the symbols/lines are labeled with 100).

The labelling error has been corrected. And the confidence interval issue is addressed by text added in response to the previous comment.

Figure 7. Although I generally agree with what is stated in the text about this figure, I am not clear how the comparisons between the three different distributions are being made here. The differences between pools, bars and riffles are not being individually discussed in terms of confidence intervals (paragraph on page 14) and you just state that the samples are or are not different. Are you somehow pooling all samples in this comparison, analogous to an ANOVA test and if so, please specify this? If not, please specify which of the distributions you mean are actually statistically different in this discussion rather than just saying samples. In Figure 7, it seems to me that most grain size percentiles could be different for the bars than for the pools or riffles, but this is not discussed until the paragraph on page 15. If bars are different from pools and riffles, then I do not understand what all of the various statistical comparisons of samples mean in the paragraph on page 14; that you state many of samples grain size percentiles are not different on page 14 but on page 15, you state that bars are different from the other two locations. I am also unclear as to why significance at 99% is now being used as the level for significance when throughout the rest of the paper, 95% was used. Why is a stricter significance level now being included?

We have re-written this section to avoid the confusion identified above. We now begin by pointing out that the bars are significantly different from the other two units (pool and run/riffle) for both streams. We then go on to compare the pools to the run/riffle units using the statistical comparision tool. We use both the 95% and the 99% confidence levels merely as an example of the sorts of comparisions that researchers might wish to make. There is no single confidence level that we recommend, and the purpose for making the comparision will determine the appropriate confidence level that a researcher might choose.

Figures 8 and 9. Why is the Bonferroni correction only applied when comparing these data and not any of the other comparisons in the paper, and what is m in these cases (each percentile compared?)?

We have added text to the manuscript which indicates that the Bonferroni correction is appropriate when comparing two distributions based on several statistical metrics, but not when comparing individual metrics, as was the case for the previous comparisons. If we wish to say this population is different from that population, then we need to use several metrics and apply the correction; if we wish to compare the D84 from this population to the D84 for another population, no correction should be applied.

P20 L9-19 Thank you for answering some of my questions about these data but I have a few remaining concerns. For example, how was visual observation of tracer movement conducted? I assume this was not during the flow that actually moved the tracers and that you simply observed that tracers were in different locations before and after a hydrograph and that the assumed critical discharge was the peak discharge value of that hydrograph? If so, can you please specify this because visual observations of tracer stone movement are usually impossible during sediment transport events and is somewhat misleading.

We are gratified that you appreciate how difficult it is to actually observe when tracers begin to move, but in this particular case, that is exactly what we did. A crew of researchers was in the field for the entire duration of the snowmelt season. Because the stream has a snowmelt generated hydrograph, the increase in flows takes place gradually over about one week. The crew was tasked with measuring discharge using a moving boat ADCP, and between discharge estimates would visit each tracer line to observe whether or not tracers had begun to move. Since the tracers were originally deployed in straight lines across the channel, were painted very bright colours, and since the stream was relatively shallow and clear up to about a discharge of about 4 m3/s. it was easy to determine when they began to move.

**1.1 Comment 1**

**2 Reviewer 3: General Comments**

The general comments from Reviewer 3 (Dr. R. Hodge) are presented below, and are discussed at some length. We attempt to address all the key issues, and highlight how we have responded to them.

**2.1 Comment 1**

One comment I had was about the assumptions made in the model. It is assumed that the probability of sampling grains larger than D50 is 0.5, with the statement that half the surface grains are smaller than D50. Both are not necessarily true. D50 is the median grain size sampled using whatever sampling technique is applied. However, if the grains are sampled on a grid, then more than half of the surface grains (by number) could be smaller than D50. The discrepancy is because larger grains are more likely to be sampled than smaller ones as they take up more space and so there is a higher probability of a grid node or foot landing on them. In the case of equal numbers of large and small grains, the larger grains would occupy more than half the surface area and be more likely to be sampled. I dont think that this sampling bias affects your analysis, but it needs more careful wording in the section around page 5, line 5. (See the Bunte and Abt 2001 technical report, section 4.3, for converting between distributions collected using different sampling techniques, e.g. grid to area.)

**2.2 reply by authors**

This is an important point that we overlooked. The wording identifed by the reviewer is innacurate, and we have changed it. Furthermore, we have added text that this approach only applies to grid-based samples or Wolman based samples, not to other kinds of samples, as the reviewer points out. For the standard Wolman (and the equivalent grid samples), the probably of picking up a particle of a certain size depends on the relative area of the bed covered by particles of that size, which is at the heart of the statistical argument for this paper.

**2.3 Comment 2**

From a quick look through the response to reviewers, it looks like the authors have worked on incorporating previous literature. There were some places where this could have been developed a bit further. For example, demonstrating the range of different recommendations that are currently in the literature. Could you have also compared the results of your bootstrapping with the findings of Rice and Church, rather than just saying that you used the same method?

**2.4 reply by the authors**

We believe that replicating the analysis used by Rice and Church is more fundamental that simply comparing our results to their empirical results. Furthermore, there is no statistical basis for making such direct comparisions in the first place. It would be like using the distribution of heights and weights for some country's Olympic swimming team to describe the distribution of heights and weights for another country's Olymic rowing team: the distributions might be similar (we are talking about Olympic atheletes), but we would expect there to be differences based on the particulars of the sport in question. So too would be expect differences in the shape of the grain size distribution curves for different rivers, based on the flow regime, sediment supply regime and local bedrock lithology. Furthermore, we are presenting a statistically based argument that overcomes the need to make this kind of flawed comparision. Therefore, we replicated the analysis, demonstrated that it is appropriate and useful, and showed that it is consistent with the predictions of binomial theory.

**3   Reviewer 3: Specific comments**

The reviewer also provides a list of specific comments that improved the paper. Those comments are quoted below, along with our responses to them. The comments are linked to page number/line number.

> 1/15: Make it clear that the spreadsheet is available with this paper?

We have added text making this explicit.

> 2/19: The problem with image based analysis is that you dont know whether you are seeing the b-axis, which could introduce a different bias into the data.

We have added text pointing out this potential bias for photographic-based methods.

> 3/3: Can you demonstrate how different the results from the different empirical analyses are, e.g. in terms of % error in D50?

How different the results are depend in large part on the population being sampled. The main point is that there is no statistical basis for applying the empirical results from one site to other sites. The text has been modified to point this out.

> 3/11: It might be helpful to summarise what Fripp and Diplas presented, e.g. the sample size suggested for a given level of precision.

We have added a summary statement to the manuscript.

> 3/17: Not clear who they is referring to.

Text added to clarify this.

> 3/28: This issue of overlapping intervals that dont include both estimates will not be unique to analysing grain size data. Why cant we use methods that have been developed by other disciplines to address it?

The methods we develop are based on existing, more widely applied statistical theory. In that sense, it is a general solution. The main problem (and the one we are trying to address with our GSDtools package) is that not many geomorphologists are using these existing, standard methods for their analyses.

> 4/13: Move bracket to before 1993.

Done.

> 5/16: It took me a couple of reads to get my head round this; it might be useful to add an additional statement (as you do later on) along the lines of In the case that 60 stones are smaller than D50, then d60 = D50.

We have added text the manuscript, along the lines suggested above.

> 7/3: After a clear overview, I wasnt sure where these subsections (2.1.1 onwards) were going. Can you add a bit more signposting? In particular, I wasnt sure what the aim of this paragraph was.

We have added a sentence indicating that the next section goes into the statistical basis for the method described in overview section, but in a more rigorous way.

> 7/10: It seemed a bit odd to be referring to a confidence interval, when you hadnt yet addressed how it was calculated.

We agree that it is a bit awkward to introduce the confidence interval here in this way. However it is discussed in the previous section, albeit without the statistical basis laid out. And, this discussion was added to emphasize the relatively small effect (relative to the size of the sample taken) of grouping data into phi size classes, an issue raised during the first round of reviews.

7/13: I needed a bit more help with comparing the two different approaches in 2.1.2 and 2.1.3. The differences seem to be that 2.1.2 gives asymmetric intervals and is mapped to specific grain size measurements. 2.1.3 seems to be symmetric, and allows interpolation between grain size measurements. When would you use the different approaches? Could you have a version that was asymmetric, but allowed interpolation?

We have modified the subsection headings to help distinguish these two approaches. The first approach is the exact solution, and strictly applies binomial theory. The key problem with the exact solution is that it produces asymmetrical confidence intervals. The second approach is an approximation based on interpolation of the binomial distribution that does produce symmetrical confidence intervals. Given the desirability of symmetrical confidence intervals, we use the approximation. The exact solution presented first because it is the basis for the approximation.

11/17: Change to measurements.

Done.

14/Fig.7: How have the polygons been calculated, i.e. which percentiles have confidence intervals been calculated for? Also, add something to the caption to explain why 22.6 mm is highlighted.

We have added text to the figure caption addressinb both these points.

14/16: Some of this explanation was a bit confusing, because the earlier analysis only refers to comparisons between two GSDs, but there are three in Fig. 7. Are all the significance values for a three-way comparison? I was surprised that the text didnt report more differences between the bars and the other two units as there seems to be almost no overlap in the figure.

This entire section has been substantially re-written. It was unclear, as written, and now we make it clear that only two units are being compared at a time (pools vs run/riffle units). We now begin the section by pointing out that, for both streams, the bars are statistically different from the other two units.

18/9: In this line and the next, clarify that you are referring to the mean . It might be helpful to give the range as well.

This has been done.

18/12: Change to wide range of.

This has been done.

[revised manuscript text omitted]